# The membrane-localized protein kinase MAP4K4/ TOT3 regulates thermomorphogenesis

Lam Dai Vu [1,2,3,4], Xiangyu Xu[1,2,10], Tingting Zhu[1,2,10], Lixia Pan[1,2], Martijn van Zanten [5], Dorrit de Jong[1,2], Yaowei Wang[1,2], Tim Vanremoortele[1,2], Anna M. Locke[6,7], Brigitte van de Cotte[1,2], Nancy De Winne[1,2], Elisabeth Stes [1,2,3,4,9], Eugenia Russinova [1,2], Geert De Jaeger [1,2], Daniël Van Damme[1,2], Cristobal Uauy [8], Kris Gevaert [3,4,11✉] & Ive De Smet [1,2,11✉]

Plants respond to mild warm temperature conditions by increased elongation growth of organs to enhance cooling capacity, in a process called thermomorphogenesis. To this date, the regulation of thermomorphogenesis has been exclusively shown to intersect with light signalling pathways. To identify regulators of thermomorphogenesis that are conserved in flowering plants, we map changes in protein phosphorylation in both dicots and monocots exposed to warm temperature. We identify MITOGEN-ACTIVATED PROTEIN KINASE KINASE KINASE KINASE4 (MAP4K4)/TARGET OF TEMPERATURE3 (TOT3) as a regulator of thermomorphogenesis that impinges on brassinosteroid signalling in *Arabidopsis thaliana*. In addition, we show that TOT3 plays a role in thermal response in wheat, a monocot crop. Altogether, the conserved thermal regulation by TOT3 expands our knowledge of thermomorphogenesis beyond the well-studied pathways and can contribute to ensuring food security under a changing climate.

[1] Department of Plant Biotechnology and Bioinformatics, Ghent University, B-9052 Ghent, Belgium. [2] VIB Center for Plant Systems Biology, B-9052 Ghent, Belgium. [3] Department of Biomolecular Medicine, Ghent University, B-9000 Ghent, Belgium. [4] VIB Center for Medical Biotechnology, B-9000 Ghent, Belgium. [5] Molecular Plant Physiology, Institute of Environmental Biology, Utrecht University, 3584CH Utrecht, The Netherlands. [6] Soybean & Nitrogen Fixation Research Unit, United States Department of Agriculture- Agricultural Research Service, Raleigh, NC 27695, USA. [7] Department of Crop and Soil Sciences, North Carolina State University, Raleigh, NC 27695, USA. [8] Department of Crop Genetics, John Innes Centre, Norwich Research Park NR4 7UH, UK. [9] Present address: VIB Headquarters, 9052 Gent, Belgium. [10] These authors contributed equally: Xiangyu Xu, Tingting Zhu. [11] These authors jointly supervised this work: Kris Gevaert, Ive De Smet. ✉email: kris.gevaert@vib-ugent.be; ive.desmet@psb.vib-ugent.be

Almost every organism is exposed to variation in temperature, on a daily and on a seasonal basis. This is especially true for plants that, as sessile organisms, need to continuously alter their growth, development and physiology in response to temperature variation[1,2]. To sense and respond to temperature changes, several molecular sensors and downstream signalling and response networks have evolved[3]. Despite that our knowledge of temperature perception and response in plants has increased in recent years, research mainly focussed on transcriptional regulation. Hence, we still know relatively little about the cellular signalling cascades that control architectural adaptations to high ambient temperatures (referred to as thermomorphogenesis). Thermomorphogenesis is characterized by traits such as upward leaf movement (thermonasty) and petiole and hypocotyl elongation in dicots, such as *Arabidopsis thaliana*, cabbages and tomato[1,2,4,5]. The resulting open rosette architecture improves the cooling capacity in unfavourable warm-temperature conditions within the physiological range[1,2,4,6,7]. In monocots, mild warm-temperatures typically affect growth rate, including leaf elongation and the length of leaf internodes[8–11].

In *A. thaliana*, the basic helix-loop-helix transcription factors PHYTOCHROME INTERACTING FACTOR 4 (PIF4) and PIF7 are central and required regulators of warm-temperature-mediated elongation growth[12–14]. Two upstream negative regulators of PIF4, namely the red light receptor phytochrome B (phyB) and the circadian clock protein EARLY FLOWERING 3 (ELF3), are thermosensors that perceive temperature information directly[15,16]. High temperature promotes the dark reversion of active nuclear-localized phyB into the cytosolic inactive form and at the same time induces reversible liquid–liquid phase separation of ELF3. In parallel, an RNA thermoswitch controls the translation of *PIF7* independently from thermosensory pathways regulating PIF4[14]. Nevertheless, both PIF4 and PIF7 dimerize and are functionally dependent on each other to regulate the transcription of a set of common genes, including auxin biosynthesis genes, such as *YUCCA8* (*YUC8*), and other auxin-responsive genes that are required for thermomorphogenesis[13,14]. In addition to phytochromes, blue light and ultraviolet-B light perceived by cryptochrome 1 (CRY1) and UV RESISTANCE LOCUS 8 (UVR8), respectively, inhibit PIF4 activity to negatively regulate thermomorphogenesis[17,18]. Altogether, PIF4 is an important signalling hub that integrates thermomorphogenesis with different light signalling components.

Considering the diverse effects of high temperature on multiple molecular components, it is likely that PIF4- and PIF7-dependent pathways are not the only thermomorphogenic pathways in Arabidopsis. Many cellular signalling events are mediated by protein phosphorylation, including plant responses to biotic and abiotic stresses[19]. For example, during response to freezing, the stability of the transcription factor INDUCER OF CBF EXPRESSION 1 (ICE1) is regulated by phosphorylation by multiple protein kinases, including a MITOGEN-ACTIVATED PROTEIN (MAP) kinase cascade. In contrast to freezing responses[20,21], little is known about phosphorylation events that regulate thermomorphogenesis. In this study, we explore the cellular high-temperature-responsive phosphorylation landscape using a phosphoproteomics approach. Hereby, we identified TARGET OF TEMPERATURE 3 (TOT3), a MITOGEN-ACTIVATED PROTEIN KINASE KINASE KINASE KINASE (MAP4K), which controls warm-temperature-responsive growth in plants in conjunction with two closely related MAP4Ks. Importantly, we demonstrate that there is a signalling pathway regulating thermomorphogenesis independently from PIF4 and light-signalling pathways, and this pathway requires TOT3 activity. We also show that TOT3 impinges on brassinosteroid-mediated growth control under warm temperature. Finally, we provide evidence that the function of TOT3 as a regulator of high-temperature-mediated growth responses is conserved in monocots.

## Results

**Phosphoproteomics reveals conserved thermoresponsive phosphorylation events.** Reversible and dynamic phosphorylation of proteins is essential for many cellular signalling cascades. We therefore analysed the phosphoproteome of *A. thaliana* Col-0 seedlings transferred from control (21 °C) to warm temperature (27 °C) (Fig. 1a). As expected, this relatively high ambient temperature induced the open rosette architecture, typical for thermomorphogenesis (Fig. 1b). To capture early and dynamic changes in protein phosphorylation status, we sampled before (0 min) and at 12, 24 and 60 min after exposure to 27 °C (Fig. 1a). This time frame aligned with the up-regulation of *HSP70* expression, a transcriptional marker associated with temperature perception status[22] (Supplementary Fig. 1). The phosphoproteome data subjected to a multiple-sample comparison between the time points revealed 212 differentially regulated phosphosites, which mapped to 180 functionally diverse proteins (Supplementary Fig. 2a and Supplementary Data 1). We manually included 14 additional phosphosites, which mapped to 12 proteins, to our selection, as these were not detected for at least one time point and thus not part of the multiple-sample comparison (Supplementary Data 1).

Next, to identify proteins involved in thermomorphogenesis and temperature perception and signalling that are regulated in both monocot and dicot plants, we compared the *A. thaliana* dataset with similar phosphoproteome datasets from wheat leaves and spikelets[23], and from soybean leaves exposed to increased temperature for 60 min (Supplementary Fig. 2b and Supplementary Data 2–4). This comparison revealed 42 putative orthologues that were also differentially phosphorylated in soybean or wheat and we named these TARGETs OF TEMPERATURE (TOTs) (Fig. 1c and Supplementary Data 1).

**TOT3 encodes a plasma membrane-localized MAP4K4.** Although temperature perception in many organisms, such as (cyano)bacteria, occurs in the plasma membrane[3,24,25], warm-temperature-sensing mechanisms residing in the plasma membrane are largely unknown in plants. We therefore prioritized TOTs that were predicted to be membrane-associated, and subsequently focused on TOT3 as the only candidate that contains a protein kinase domain (Fig. 1c, Supplementary Fig. 4a and Supplementary Data 1). *TOT3* encodes MAP4K4, a conserved member of a clade of protein kinases referred to as the MAP4K family[26] (Supplementary Fig. 3). The MAP4K family is evolutionarily conserved and related to yeast Ste20[27], and MAP4Ks can activate MAPK cascades or phosphorylate diverse substrates[28]. In Arabidopsis, BLUE LIGHT SIGNALLING1 (BLUS1)/MAP4K10 functions downstream of phototropins to control stomatal opening[29]. In addition, both TOT3/MAP4K4 and SIK1/MAP4K3 were shown to play a role in ROS production during plant immunity response[30,31]. The expression of *TOT3* remained stable during a warm-temperature time course (Supplementary Fig. 4b), indicating that regulation of the transcription of *TOT3* cannot explain its effect on thermomorphogenesis. Next, we confirmed that at least a part of GFP:TOT3 is indeed plasma membrane-localized in the hypocotyl, in cotyledons and in the primary root meristem (Fig. 1d, e and Supplementary Figs. 5 and 6).

**TOT3 is important for thermoresponsive growth in plants.** Next, we genetically explored a role for TOT3 in thermoresponsive growth in plants. Two independent *A. thaliana* mutant lines carrying alleles lacking full-length *TOT3* transcript (*tot3-1* and *tot3-2*) displayed a significantly shorter hypocotyl at 28 °C when compared to Col-0; but showed no significant differences at 21 °C (Fig. 2a, b and Supplementary Figs. 7 and 8). In

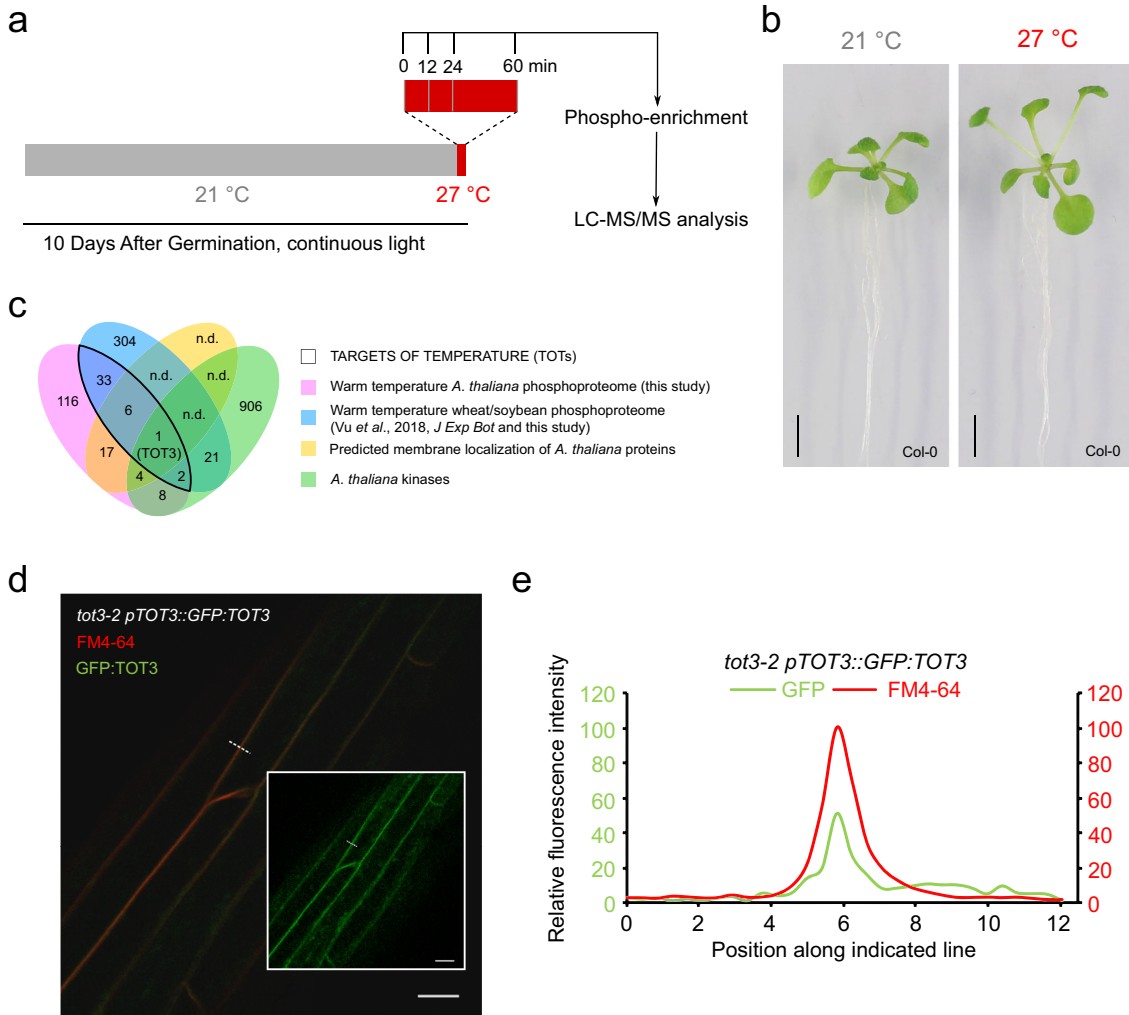

**Fig. 1 Phosphoproteome profiling reveals TOT3/MAP4K4 as a potential regulator of thermomorphogenesis in Arabidopsis. a** Schematic overview of the experimental setup. **b** Representative seedlings subjected to the experimental setup, after 3 days of exposure to 27 °C. Scale bar, 5 mm. **c** Venn diagram comparing datasets to define TARGETs OF TEMPERATURE (TOTs), and specifically TOT3. n.d., not determined (plasma membrane localization only predicted for candidates from *A. thaliana* experiment). Kinases based on ref. [73]. **d, e** Merged red and green signals of localization of GFP:TOT3 (green) in the etiolated hypocotyl of *tot3-2* seedlings expressing *pTOT3::GFP:TOT3* seedlings grown at 28 °C and stained with the endocytic tracer dye FM4-64 (red) (**d**). Inset in (**d**) shows GFP:TOT3 signal alone. Quantitation of relative fluorescence intensity at 493-532 nm (GFP) and 596-645 (FM4-64) along dotted white line (**e**). Scale bar, 20 μm (**d**).

contrast, we did not detect any obvious differences with respect to flowering time, another hallmark of warm-temperature responsiveness[2], at 21 or 28 °C (Supplementary Fig. 9). A *pTOT3::GFP:TOT3* construct complemented the *tot3-2* hypocotyl phenotype at 28 °C (Fig. 2c, d), confirming that TOT3 activity is required for warm-temperature-mediated elongation and that the GFP:TOT3 fusion is functional.

Since we also identified differential phosphorylation of a TOT3 orthologue in wheat[23] (Fig. 1c, Supplementary Fig. 10 and Supplementary Data 4), we investigated whether regulation of thermomorphogenesis by TOT3 is functionally conserved. We identified Cadenza wheat TILLING lines[32] with a premature stop codon in the coding sequence (CDS) of the three wheat *TOT3* homeologues (Fig. 3a). Little is known about thermomorphogenesis in monocots[8–11], but wild-type Cadenza wheat seedlings displayed high-temperature-triggered growth promotion of the leaf sheath when grown at 24 °C compared to 14 °C (Supplementary Fig. 11a). Interestingly, Cadenza wheat TILLING lines with mutations in the wheat *TOT3* gene *TraesCS7D02G232400* showed a significantly shorter second leaf sheath at 24 °C compared to 14 °C (Fig. 3b, c),

whereas those with mutations in the homeologous *TraesC-S7A02G232300* and *TraesCS7B02G130700* genes did not show any significant difference (Supplementary Fig. 11b, c). In contrast, a temperature increase up to 34 °C repressed wheat seedling growth (Supplementary Fig. 12). However, *tot3* mutant wheat seedlings were more stunted than wild-type plants at 34 °C (Supplementary Fig. 12). Taken together, our data support that TOT3 is a conserved regulator of thermoresponsive growth in plants.

**Thermomorphogenesis in Arabidopsis requires TOT3 kinase activity.** Since some MAP4Ks can act as molecular adaptors rather than as bona fide kinases[28], we evaluated if the TOT3 kinase domain and its kinase activity are required for its function. TOT3 protein variants that lacked the kinase domain (3xHA: TOT3$^{309-674}$) or in which aspartate 157 in the conserved DFG motif of the kinase domain was replaced by asparagine (GFP: TOT3$^{D157N}$), which abolishes kinase activity[29], could not rescue the *tot3-2* short hypocotyl phenotype at 28 °C (Supplementary Figs. 13 and 14a, b). In addition, MBP-TOT3WT-6xHIS

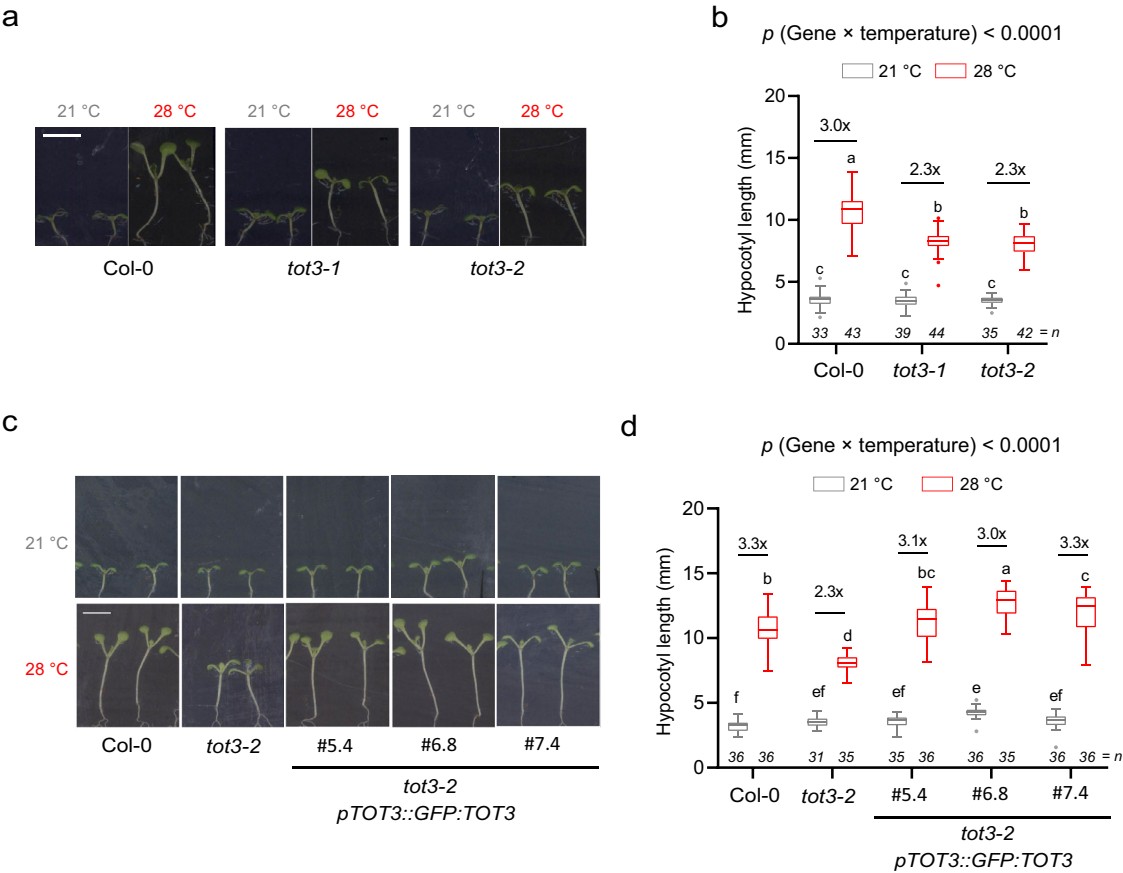

**Fig. 2 TOT3 is required for thermomorphogenesis in Arabidopsis. a, b** Hypocotyl length of 7-day-old Col-0 wild-type and loss-of-function *tot3-1*, *tot3-2* under short-day conditions at 21 and 28 °C ($33 \leq n \leq 44$). **c, d** Hypocotyl length of 7-day-old *tot3-2* plants complemented with *pTOT3::GFP:TOT3* (three independently transformed lines: #5.4, #6.8 and #7.4) under short-day conditions at 21 and 28 °C. Representative pictures (**a, c**) and hypocotyl length quantification (**b, d**). Scale bar, 5 mm. Box plots show median with Tukey-based whiskers and outliers. The number of individually measured seedlings (*n*) is indicated above the *X*-axis. Fold-change is indicated for each genotype. Letters indicate significant differences based on two-way ANOVA and Tukey's test ($p < 0.01$). The *p*-value for the interaction (genotype × temperature) is shown at the top.

recombinantly expressed in *Escherichia coli* showed autophosphorylation in vitro while MBP-TOT3[D157N]-6xHIS did not, indicating that TOT3 is an active kinase and that the D157N mutation abolishes its activity (Supplementary Fig. 14c). This supports that the TOT3 kinase domain and activity are indeed required for TOT3-mediated growth at warm temperature.

**TOT3 regulates thermomorphogenesis independently of phyB and PIF4.** Next, we evaluated the genetic interaction between TOT3 and known temperature-signalling components, and hypocotyl growth regulators phyB and PIF4[1,4]. While *tot3-2* and *phyb-9* displayed a short and long hypocotyl compared to Col-0 at 28 °C, respectively, the *tot3-2 phyb-9* double mutant has an intermediate phenotype (Fig. 4a, b), suggesting that TOT3 and phyB signalling have additive effects on thermomorphogenesis. On the contrary, the *pif4-101 tot3-2* double mutant showed significantly less increase in hypocotyl length than either single mutant at 28 °C, pointing to a possible additive effect of both factors (Fig. 4c, d). Next, the expression of a core set of temperature-responsive genes, including *PIF4* and its two downstream target genes *YUC8* and *ATHB2*, which are involved in thermomorphogenesis and act downstream of phyB[4], was assessed. Although some of these genes displayed a slightly lower expression level in *tot3-2*, their temperature-triggered up-regulation was not significantly affected (Fig. 4e). The observation

that transcript levels of *YUC8* (a rate limiting enzyme in warm-temperature-induced auxin biosynthesis) and of *IAA29* (an auxin response gene involved in hypocotyl growth) were not or hardly affected in the *tot3-2* mutant background (Fig. 4e and Supplementary Fig. 15a) suggests that auxin responses are largely not affected by TOT3. This was confirmed by pharmacological application of picloram, a synthetic auxin that promotes hypocotyl growth in Col-0[33]. Picloram triggered hypocotyl elongation in *tot3-2* to a comparable extent as in Col-0 at control temperature (21 °C) (Supplementary Fig. 15b), further indicating that auxin signalling is not perturbed in the *tot3-2* mutant.

Both phyB and PIF4 are light signalling components that are, in addition to thermomorphogenesis, also involved in other responses, such as shade avoidance[34,35]. The observation that *tot3* hypocotyl length at 21 °C is similar to that of Col-0 (Fig. 2a, b) may indicate that TOT3 is not required for hypocotyl growth in the absence of a warm-temperature cue. Indeed, hypocotyl length of *tot3* at 21 °C in the presence of a low red-to-far-red (R/Fr) light ratio, a typical canopy shade signal that leads to elongated hypocotyls[34,35], is similar to wild-type (Supplementary Fig. 15c). Next, we examined thermoresponsive hypocotyl growth in darkness at 21 and 28 °C. During etiolation, the activity of PIFs should be maximized due to the lack of active phytochromes in the absence of light. Strikingly, in this setup, we observed a similar increase in hypocotyl length of both *pif4-101* and Col-0 at 28 °C, compared to 21 °C in darkness (Fig. 4f, g). In contrast, hypocotyl

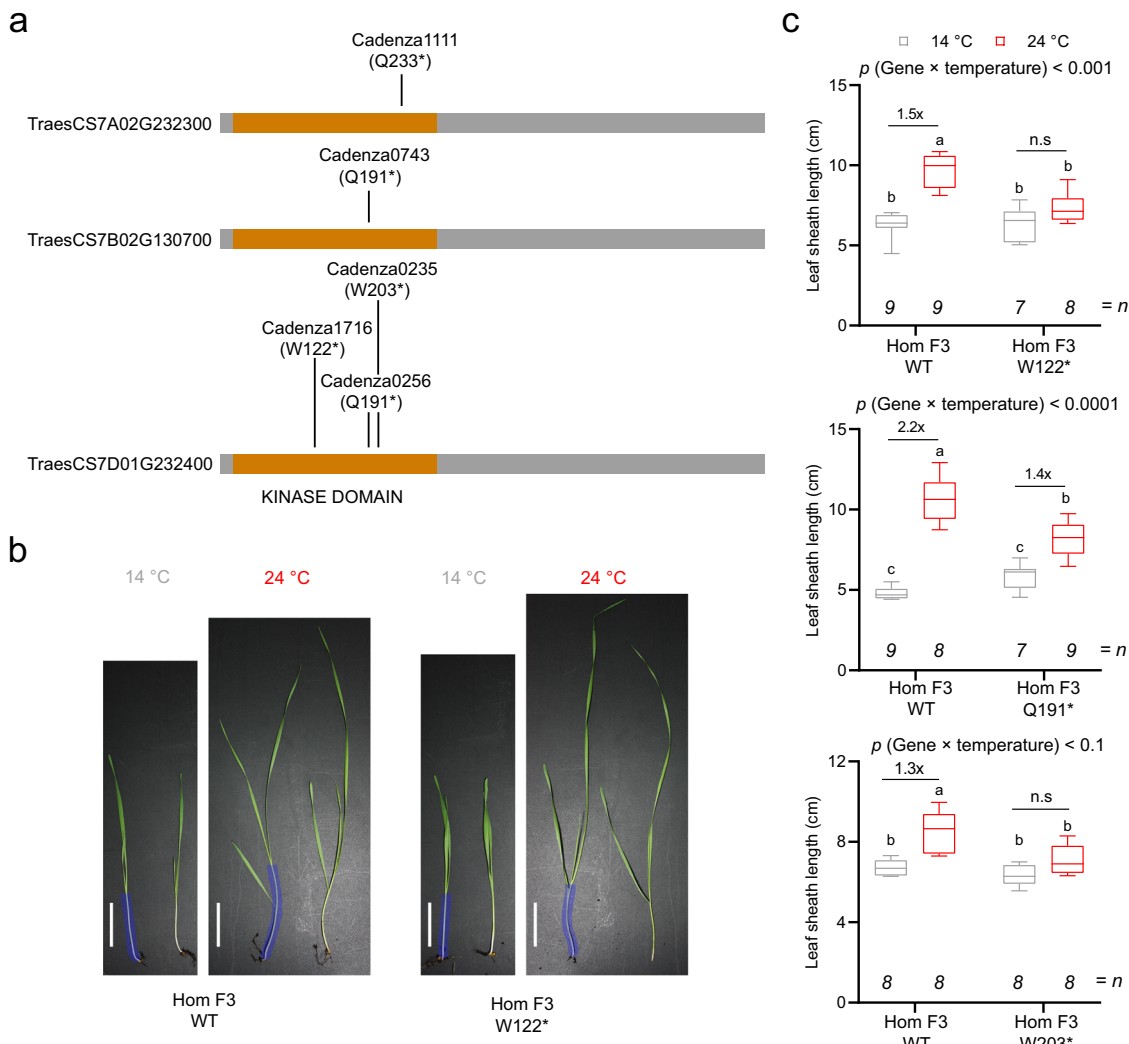

**Fig. 3 TOT3 is required for thermomorphogenesis in wheat. a** Position of stop codons (*) in wheat TILLING lines Cadenza1111 (Q233*), Cadenza0743 (Q191*), Cadenza1716 (W122*), Cadenza0256 (Q191*) and Cadenza0235 (W203*) in protein models for three wheat TOT3 homeologues. **b, c** Length of the visible leaf sheath (in those cases where the second leaf has not emerged, this leaf sheath is underneath the one of the first leaf) in homozygous F3 wild-type (WT) and *tot3* two-week old Cadenza wheat seedlings (all lines were backcrossed once with WT Cadenza and WT plants were selected from the backcrossing with each Cadenza TILLING line) grown at 14 and 24 °C in long-day conditions (16 h light/ 8 h dark). Representative pictures (**b**) and leaf sheath length quantification (**c**). Scale bar, 2 cm. Blue overlay in (**b**) marks the leaf sheath. Box plots show median with Tukey-based whiskers and outliers. The number of individually measured seedlings (*n*) is indicated above the *X*-axis. The respective quantification letters indicate significant differences based on two-way ANOVA and Tukey's test ($p < 0.01$); n.s: not significant (**c**). The *p*-value for the interaction (genotype × temperature) is shown at the top.

length of *tot3-2* seedlings remained the same at both 21 and 28 °C in darkness (Fig. 4f, g and Supplementary Fig. 15d). Furthermore, *pif4-101 tot3-2* seedlings exhibited a similar phenotype as the *tot3-2* mutant. This observation agrees with the fact that the transcript levels of *PIF4* and its targets were not increased at 28 °C compared to 21 °C in both Col-0 and *tot3-2* plants in darkness (Fig. 4h). This contrasts with what has been routinely observed in other light regimes[36] (Fig. 4e). In conclusion, our results indicate that TOT3 defines a temperature-signalling pathway controlling hypocotyl thermomorphogenesis independent from the light signalling components phyB and PIF4.

**TOT3 interacts with MAP4K6/TOI4 and MAP4K5/TOI5 to regulate thermomorphogenesis.** Next, we explored (direct) downstream targets and interacting proteins of TOT3, to position TOT3 in a cellular signalling cascade triggering warm-temperature-induced growth. We therefore performed a

protein–protein interaction study using tandem affinity purification (TAP) on GS^rhino-tagged TOT3 (GS^rhino:TOT3) expressed in an *A. thaliana* cell culture. The interactome revealed 23 TOT3-INTERACTING proteins (TOIs), including MAP4K5, MAP4K6, MAP4K7, MAP4K8 and MAP4K9, which are all MAPK4Ks closely related to TOT3 (Fig. 5a, Supplementary Fig. 3 and Supplementary Data 4). A subsequent co-immunoprecipitation experiment using *pTOT3::GFP:TOT3*-expressing *tot3-2* seedlings confirmed the interactions between TOT3 and TOI4/MAP4K6 or TOI5/MAP4K5 and showed that these interactions were temperature-independent (Fig. 5a and Supplementary Data 5). The TOT3–TOI4/5 interaction was further confirmed by yeast two-hybrid (Y2H) analyses (Fig. 5b). Although stress-related gene ontology terms were enriched among potential interactors identified in the TAP and GFP pull-down experiments (Supplementary Data 5), the above-described protein–protein interaction datasets did not identify an

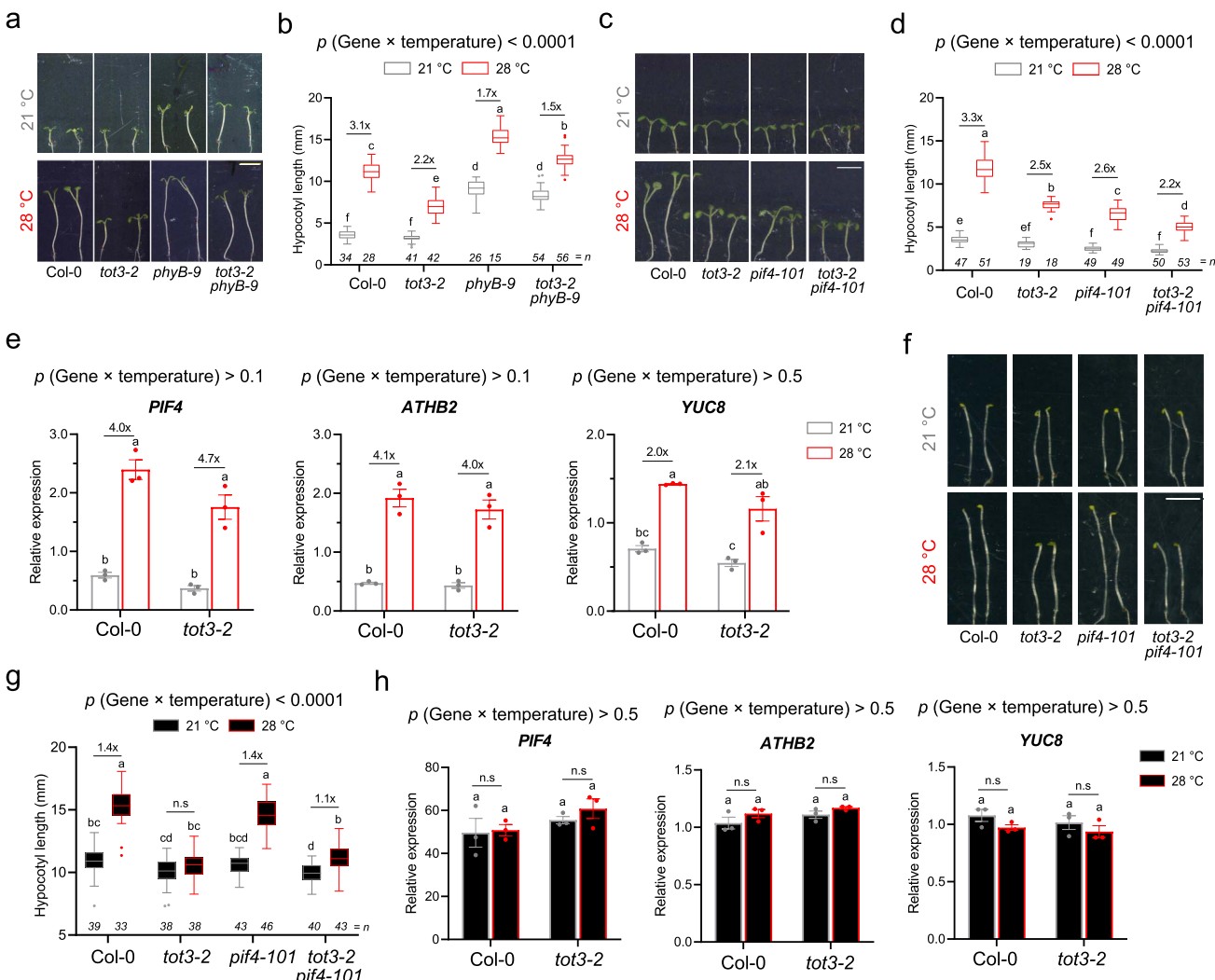

**Fig. 4 TOT3 regulates thermomorphogenesis independently from light signalling. a–d** Genetic interaction between 7-day-old *tot3-2* and *phyb-9* (**a**, **b**) ($15 \leq n \leq 54$) and *pif4-101* (**c**, **d**) ($18 \leq n \leq 53$) at 21 and 28 °C under short-day conditions. Representative pictures (**a**, **c**) and hypocotyl length quantification (**b**, **d**). Scale bar, 5 mm (**a**, **c**). **e** Relative expression of temperature-responsive genes in 3-day-old Col-0 wild-type and *tot3-2* seedlings in control (21 °C) and warm-temperature conditions (28 °C) under short-day conditions. **f**, **g** Genetic interaction between 3-day-old *tot3-2* and *pif4-101* at 21 and 28 °C in darkness. Representative pictures (**f**) and hypocotyl length quantification (**g**). Scale bar, 5 mm (**f**). **h** Relative expression of temperature-responsive genes in 3-day-old Col-0 wild-type and *tot3-2* seedlings in control (21 °C) and warm-temperature conditions (28 °C) in darkness. Box plots show median with Tukey-based whiskers and outliers (**b**, **d**, **g**). The number of individually measured seedlings (*n*) is indicated above the *X*-axis (**b**, **d**, **g**). Bar diagram shows mean of 3 biological replicates (individual dots) with standard error of the mean (**e**, **h**). For the quantitative figures, letters indicate significant differences based on two-way ANOVA and Tukey's test ($p < 0.01$) with fold-change between 21 and 28 °C being presented; n.s: not significant. The *p*-value for the interaction (genotype × temperature) is shown at the top.

obvious signature associating it with known (temperature) signalling pathways. Nevertheless, a role for TOI4 and TOI5 in TOT3-mediated warm-temperature signalling is indirectly supported by phosphoproteome data, which revealed that TOI4 and TOI5 phosphopeptides were less abundant in *tot3-2* at 28 °C compared to Col-0 (Fig. 5a, c and Supplementary Data 6). Furthermore, warm-temperature-induced changes in phosphorylation status of TOI4 and TOI5 orthologues were also observed in soybean and wheat[23] (Supplementary Fig. 17). Although TOI4 and TOI5 expression levels are not affected in *tot3* (Supplementary Fig. 16), we can however, not rule out that TOI4 and TOI5 protein levels might be affected in *tot3-2* and/or that the stability of TOI4 and TOI5 in a complex might be higher than that of the individual proteins.

Next, to further explore TOT3-mediated phosphorylation of TOI4/5, we transiently co-expressed the kinase-dead RFP-

TOI4[D188N] and RFP-TOI5[D174N] with wild-type TOT3 or the kinase-dead TOT3[D157N] in tobacco leaves and analysed the phosphorylation status of the immunoprecipitated RFP-TOI4[D188N] and RFP-TOI5[D174N]. An increased phosphorylation at S499, S516 and S654 on TOI5[D174N] was observed when co-expressed with wild-type TOT3, compared to the kinase-dead TOT3[D157N] (Supplementary Fig. 18 and Supplementary Data 7). However, no differences in phosphorylation were observed for the kinase-dead TOI4[D188N] (Supplementary Fig. 18 and Supplementary Data 6). In addition, in vitro phosphorylation analysis of recombinant kinase-dead TOI4[D188N] or TOI5[D174N] proteins showed no difference between incubation with wild-type TOT3 or with kinase-dead TOT3[D157N] (Supplementary Data 8). This suggests that TOT3, at least in vitro, does not directly phosphorylate TOI4 and TOI5, and that phosphorylation of TOI4 and TOI5—although both interact with TOT3—is likely

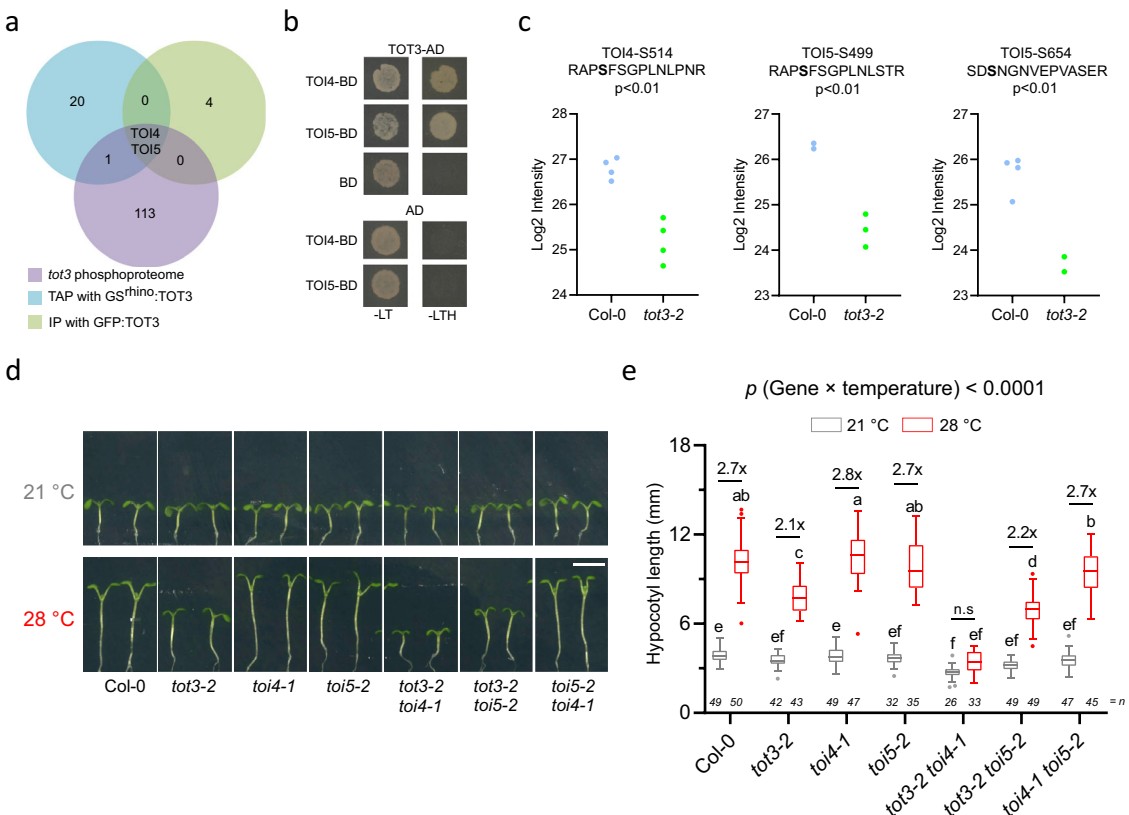

**Fig. 5 Related MAP4Ks that interact with TOT3 play a role in regulating hypocotyl growth under warm temperature. a** Overlap between TOT3 interactome (based on tandem affinity purification (TAP) and immunoprecipitation (IP)) and differentially phosphorylated proteins in *tot3-2* versus Col-0 (referred to as *tot3* phosphoproteome obtained from 3-day-old *tot3-2* and Col-0 seedlings grown at 28 °C). **b** Yeast two-hybrid validation of the interaction between TOT3 and TOI4/5. BD, BINDING DOMAIN; AD, ACTIVATION DOMAIN; -LT, without Leu and Trp; -LTH, without Leu, Trp and His. **c** TOI4/5 phosphosites significantly deregulated in *tot3* phosphoproteome. Each dot represents a biological replicate. Statistical analyses using Student's *t*-test with two-sample unequal variance and two-tailed distribution; *p*-value is indicated. **d**, **e** Hypocotyl length of 7-day-old *tot3*, *tot3 toi4*, *tot3 toi5*, *toi4 toi5* and Col-0 plants at 21 and 28 °C in short-day conditions. Representative pictures (**d**) and hypocotyl length quantification (**e**). Scale bar, 5 mm. Box plots show median with Tukey-based whiskers and outliers. The number of individually measured seedlings (*n*) is indicated above the *X*-axis. Fold-change is indicated for each genotype; and letters indicate significant differences based on two-way ANOVA and Tukey's test (*p* < 0.01) (**e**). The *p*-value for the interaction (genotype × temperature) is shown at the top.

controlled *in planta* by an unknown regulator. Future analyses will have to reveal the role of the direct interaction between TOT3 and TOI4/5 in thermoresponsive growth.

To test for an anticipated role for TOI4 and TOI5 in temperature signalling, we generated loss-of-function *toi4* and *toi5* mutants (Supplementary Fig. 19), as well as various double-mutant combinations with *tot3*. Both *toi4-1* and *toi5-2*, as well as the *toi4-1 toi5-2* double mutant, displayed similar warm-temperature-induced hypocotyl elongation at 28 °C under short-day conditions as Col-0 (Fig. 5d, e). Furthermore, *tot3-2 toi5-2* plants exhibited only a slight difference in thermormorphogenic hypocotyl length compared to *tot3-2* plants, whereas the response is largely abolished in *tot3-2 toi4-1* plants (Fig. 5d, e). These observations seem to emphasize a dominant genetic role of *TOT3* in the thermomorphogenic pathway, probably in conjunction with *TOI4*, whereas *TOI5* may play a minor role. However, when grown in darkness at 28 °C, *toi4-1 toi5-2* had a slightly shorter hypocotyl than the wild-type (Supplementary Fig. 20). Since *tot3-2* has an even shorter hypocotyl when grown in darkness at 28 °C (Supplementary Fig. 20), there is possibly genetic redundancy with other MAP4Ks or, alternatively, other TOIs are involved. Nevertheless, our data position TOT3 as a central regulator of temperature-mediated growth.

**TOT3 impinges on brassinosteroid signalling**. We also observed that at 28 °C several brassinosteroid signalling components[37–39] were less phosphorylated in *tot3-2* compared to Col-0 (Supplementary Fig. 21). Given the prominent role of brassinosteroid signalling in controlling thermomorphogenesis[40,41], we explored a possible link between TOT3 and this phytohormone. Despite the lower phosphorylation levels of several upstream brassinosteroid signalling components, such as BSU1, BSK7 and BSK8, *tot3-2* did not affect the brassinolide-induced accumulation of non-phosphorylated BES1, a hallmark for brassinosteroid signalling activation[42], at 28 °C compared to wild-type (Fig. 6a and Supplementary Fig. 22). Treatment with brassinolide increases hypocotyl length at 21 °C similarly in Col-0 and *tot3-2* (Fig. 6b). In our hands, however, the same brassinolide treatment resulted in a shorter hypocotyl at 28 °C (Fig. 6b), in contrast to previously reported results[40]. This can be due to differences in the experimental setup and/or the different source or concentration of brassinolide (see "Methods"), but both growth promoting and repressing activities are not uncommon for a hormone[43]. Nevertheless, a strong reduction in hypocotyl length in the presence of brassinolide at 28 °C was not observed in *tot3-2* plants (Fig. 6b), suggesting that TOT3 might impact brassinosteroid signalling.

We next focused on BZR1, one of the major transcription factors regulated by brassinosteroid signalling that is required for thermomorphogenesis[40]. The *bzr1-1D* gain-of-function mutant

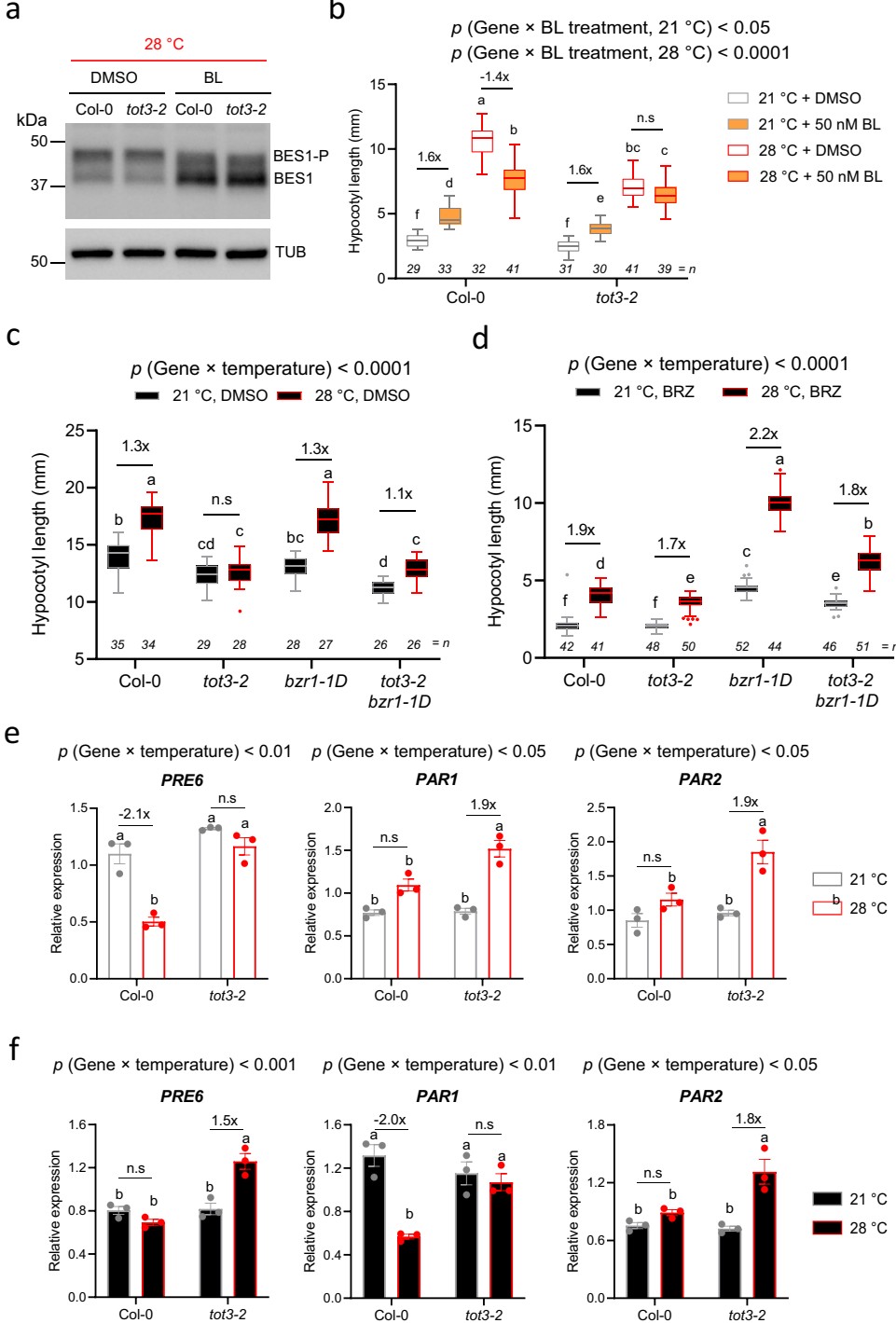

**Fig. 6 TOT3 impinges brassinosteroid signalling pathway. a** Representative western blot (of 3 biological replicates) of BES1 phosphorylation status in 4-day-old Col-0 and *tot3-2* seedlings grown on mock (DMSO) or 100 nM brassinolide (BL) at 28 °C under short-day conditions. BES1 was detected with anti-BES1 antibody. BES1-P, phosphorylated BES1; BES1, dephosphorylated BES1. Detection of TUBULIN (TUB) was used as a loading control. **b** Hypocotyl length of 5-day old *tot3-2* and Col-0 wild-type plants grown at 21 and 28 °C in short-day conditions on medium containing DMSO or 50 nM brassinolide. **c**, **d** Hypocotyl length of 4-day old *tot3-2*, *bzr1-1D*, *tot3-2 bzr1-1D* and Col-0 wild-type seedlings grown at 21 and 28 °C in darkness on MS/2 medium containing DMSO (**c**) or 1 μM brassinazole (**d**). **e** Relative expression of BZR1 target genes *PRE6*, *PAR1* and *PAR2* in Col-0 and *tot3-2* plants grown at control (21 °C) and warm temperature (28 °C) under short-day conditions. **f** Relative expression of BZR1 target genes *PRE6*, *PAR1* and *PAR2* in Col-0 wild-type and *tot3-2* plants grown at control (21 °C) and warm temperature (28 °C) under dark conditions. Box plots show median with Tukey-based whiskers and outliers (**b**, **c**, **d**). The number of individually measured seedlings (*n*) is indicated above the *X*-axis (**b**, **c**, **d**). Bar diagram shows mean of 3 biological replicates (individual dots) with standard error of the mean (**e**, **f**). Letters indicate significant differences based on two-way ANOVA and Tukey's test ($p < 0.01$); n.s: not significant. The *p*-value for the interaction (genotype × temperature or genotype × treatment) is shown at the top.

largely suppresses brassinosteroid-deficient and brassinosteroid-insensitive phenotypes and shows a BR hyperresponsive phenotype, for example, during etiolation in darkness[44,45]. However, the bzr1-1D mutation was not able to rescue the short tot3-2 hypocotyl in the tot3-2 bzr1-1D double mutant at 28 °C in darkness (Fig. 6c). Treatment with brassinazole, an inhibitor of brassinosteroid biosynthesis, largely suppressed hypocotyl growth of both Col-0 and tot3-2 seedlings at both 21 and 28 °C in darkness (Fig. 6d). However, while the bzr1-1D mutant largely rescued this phenotype, the effect of bzr1-1D is significantly dampened in the tot3-2 bzr1-1D mutant (Fig. 6d). Taken together, these results suggest that TOT3 affects BZR1-mediated brassinosteroid responses, possibly through regulating active BZR1.

To further explore this possible explanation, we assessed the expression of genes that are directly regulated by BZR1 in the tot3-2 mutant: PACLOBUTRAZOL RESISTANCE1 (PRE1), PRE5, PRE6, PHY RAPIDLY REGULATED 1 (PAR1) and PAR2[46,47]. In nearly all cases, the 28 °C-mediated change in transcript levels is different in tot3-2 compared to Col-0 under short-day conditions or in darkness (Fig. 6e, f and Supplementary Fig. 23). Considering the cell expansion-promoting effect of PREs and the cell expansion-repressing effect of PAR1/2[48,49], the different expression profiles of these genes at 28 °C cannot fully explain the reduced hypocotyl length in tot3-2. However, the reprogramming of their expression in tot3-2 mutants again points to altered activity of their upstream regulators, including BZR1, and suggests that the impact on the gain-of-function bzr1-1D is representative for the endogenous role of TOT3 on BZR1.

Finally, the tot3-2 toi4-2 toi5-1 triple mutant exhibited a striking dwarf phenotype that resembles the phenotype of bri1-116, a null mutant for the brassinosteroid receptor BRASSINOSTEROID INSENSITIVE1 (BRI1)[50] (Supplementary Fig. 24), suggesting that both brassinosteroid signalling and TOT3 pathways may affect the same growth-controlling component(s).

In conclusion, these results indicate that TOT3 controls brassinosteroid-mediated hypocotyl growth under warm temperature in darkness, through gating BZR1 activity.

## Discussion

Plants incorporate temperature information into their life cycles in several ways. Therefore, climate warming strongly impacts plant growth and yield[1,2,51]. Here, we present the membrane-associated protein kinase TOT3/MAP4K4 as a new key player in plant responses to warm temperature. In Arabidopsis, thermoresponsive hypocotyl growth intersects with light-signalling pathways and is mainly controlled by the transcription factors PIF4 and PIF7 and their (co-)regulators[1]. Considering temperature is a non-ligand environmental stimulus that affects every cellular component, it is unlikely that there is only a single regulatory pathway that controls thermomorphogenesis. Indeed, pif4 mutants grown in a normal dark/light regime often show a partial thermoresponsive hypocotyl elongation[36,52] (Fig. 4c, d). Additionally, about 21% of the warm-temperature-regulated transcriptome is not deregulated in phyABCDE quintuple null mutant plants[15]. Here, we propose a thermomorphogenesis pathway governed by TOT3 that operates independently of phyB and PIF4 (Supplementary Fig. 25). This is supported by our observation that in the dark, where activity of PIFs reaches the maximum due to the lack of active phytochromes, both Col-0 and pif4-101 hypocotyl still elongated at 28 °C, whereas this response is abolished in both tot3-2 and tot3-2 pif4-101 mutants. Moreover, the intermediate hypocotyl length of the tot3-2 phyb-9 double-mutant compared to the respective single-mutant plants further strengthens our model.

While the MAP kinase signalling cascades in plants have been extensively studied in the context of development and stress responses, not much is known about the function of plant MAP4Ks and whether these proteins function independently from the canonical MAP3K-MAP2K-MAPK cascades[26]. Here, we show that MAP4K4/TOT3 interacts with related MAP4Ks (MAP4K6/TOI4 and MAP4K5/TOI5) and that these related kinases also play a role in thermomorphogenesis in conjunction with TOT3 to varying extents. In plants, interaction of closely related kinases has been observed for several receptor-like kinases, which act together in ligand perception and downstream phosphorylation[53]. Another well-known example is OPEN STOMATA 1 (OST1)/SUCROSE NON-FERMENTING RELATED KINASE 2.6 (SnRK2.6), which forms a complex with SnRK2.2, SnRK2.3 and SnRK2.8, potentially amplifying SnRK2 signalling upon salt stress or osmotic stress[54]. Of note, while these kinases show significant functional redundancy, they also exhibit differences in their function and regulation[55]. Similarly, homo- and heterodimer formation is also observed for mammalian MAP4Ks, while they also have distinct functions and interactors[56]. The thermomorphogenic phenotype of tot3 toi4 and tot3 toi5 double mutants, and especially the phenotype of the tot3 toi4 toi5 triple mutant, suggest the existence of functional redundancy between TOT3, TOI4 and TOI5 with respect to particular phenotypes. However, the toi4 toi5 double-mutant phenotype does not exhibit an obvious phenotype at high temperature, indicating that TOT3 plays a dominant (genetic) role in the response. Alternatively, the role of TOI5 might largely overlap with TOT3, as both interact and TOI5 phosphorylation depends on TOT3, while TOI4 appears to play a largely redundant or alternative role. Given their interaction, future work should address if these MAP4Ks function in the same complex in planta and to what extent (de)phosphorylation events attribute to relaying the temperature signal to downstream signalling components. For example, in our Arabidopsis phosphoproteome dataset, the serine 333 (S333) of TOT3 shows significant increased phosphorylation at 60 min. This residue is highly conserved among MAP4Ks and also among TOT3 homologues in other land plants (Supplementary Fig. 26), indicating a possibly conserved functional importance for this phosphosite. Similarly, the phosphorylation of S348 in BLUS1 is triggered by blue light and is important for its activity[29]. In addition, we observed indirect, but TOT3-dependent phosphorylation of TOI4 and TOI5 that needs to be explored in the future.

TOT3 is potentially required for conveying the stimulating effect of brassinosteroids on warm-temperature-triggered hypocotyl elongation in darkness (Fig. 6), whereas a link with auxin is less prominent (Fig. 4 and Supplementary Fig. 15). Our data suggest that brassinosteroid signalling might be severely impaired in the tot3-2 toi4-1 toi5-2 mutant. However, it remains to be investigated how TOT3 controls BZR1 activity and how the output of the TOT3 complex with TOI4 and TOI5 affects growth, and particularly its interactions with light signalling networks need to be considered (Supplementary Fig. 26), especially since TOT3 seemingly acts independent of shade avoidance and parallel to PIF4-mediated thermosignalling (Fig. 4).

Finally, while elongation of aerial organs in dicots is a mechanism to cope with high temperature deviating from the optimal temperature, in wheat the role of high-temperature-mediated growth responses is not well studied. Nevertheless, TOT3 is also involved in regulating growth responses in wheat at temperatures above the optimal growth temperature (Fig. 3 and Supplementary Fig. 12). While we indeed only found differential phosphorylation of TOT3 orthologues in wheat spikelets, we also found TOI4 and TOI5 orthologues in the soybean and wheat leaf data. This suggests that components of the TOT3 signalosome are conserved and may also be involved in high temperature responses in these species. However, given that mass

spectrometry does not give a full proteome-wide overview in every independent analysis, the lack of identifying TOT3 phosphorylation does not necessarily indicate that it does not occur in wheat leaves. Thus, the conserved TOT3 complex has great potential for knowledge-based breeding of warm-temperature-resilient crops that can contribute to upkeep future food security in a warming climate.

## Methods

**Plant materials and growth conditions**. All *A. thaliana* plants used in this study were in the Col-0 reference accession genetic background and referred to as wild-type. The following *A. thaliana* lines were used: *tot3-1* (SALK_065417), *tot3-2* (SALK_086087), *phyb-9* (ref. [57]), *pif4-101* (ref. [58]), *bzr1-1D* (ref. [44]). For soybean phosphoproteome profiling, we used *Glycine max* [L.] Merr. cv. "Benning HP" (ref. [59]). Arabidopsis seeds were sown on Murashige and Skoog (MS) growth medium containing 1% sucrose (per litre: 2.15 g of MS salts, 0.1 g of myo-inositol, 0.5 g of MES, 10 g of sucrose and 8 g of plant tissue culture agar; pH 5.7). For treatment with picloram (Sigma-Aldrich, CAS: 1918-02-1) and brassinolide (OlchemIm, Catalogue Nr.: 015 5893), MS medium was supplemented with the concentration indicated in the text. For phosphoproteome profiling of Col-0 plants at high temperature, Col-0 seeds were sown on horizontal plates. After germination, plates were transferred to continuous light, 21 °C for 10 days before undergoing temperature treatment at 27 °C (100 µmol m$^{-2}$ s$^{-1}$ photosynthetically active radiation supplied by cool-white, fluorescent tungsten tubes, Osram). For comparative phosphoproteomics of Col-0 and *tot3-2* plants, after seed germination, plates were transferred to short-day condition at 28 °C for 3 days. Samples were collected at the end of the third night. For soybean (Glycine max [L.] Merr. cv. "Benning HP" 4) phosphoproteome profiling, seeds were germinated in ragdolls for 2 days under dark conditions and subsequently transplanted to soil in growth chambers. Plants were grown at 30 °C day/20 °C night, with a 9 h photoperiod and 2 h darkness. Daytime photosynthetic photon flux density in the growth chambers was approximately 700 µmol m$^{-2}$ s$^{-1}$. The 2 h darkness interruption was removed 24 days after transplanting to induce reproductive development. When plants were in full flower (~38 days after transplanting), the temperature was increased to 35 °C for 1 h, before the whole leaflets from the uppermost fully expanded leaf were excised and flash-frozen for phosphoproteomics. Four plants were sampled per temperature treatment. For cotyledon images of GFP:TOT3 localization, after germination in the continuous light growth chamber at 21 °C, plates were wrapped plates in aluminium foil to prevent formation of the autofluorescence chlorophyll. Seedlings were then either left at 21 °C or put at 28 °C where they remained for an additional 3 days prior to imaging.

**Wheat TILLING lines**. The wheat *TOT3* ortholog ID in the D genome (TraesCS7D02G232400) was searched on the EnsemblPlants website (plants.ensembl.org/Triticum_aestivum/Transcript/Variation_Transcript/Table?db=core;g = TraesCS7D02G232400;r = 7D:193922055-193930244;t = TraesCS7D02G232400.2) to find TILLING mutations (www.wheat-training.com). The wheat *tot3* TILLING lines[32,60], Cadenza1111 (Q233*), Cadenza0743 (Q191*), Cadenza1716 (W122*), Cadenza0256 (Q191*) and Cadenza0235 (W203*), were ordered directly through the UK Germplasm Resource Unit website (www.seedstor.ac.uk). The seeds were sown into soil in the greenhouse and cultivated at 24 °C day (16 h)/18 °C night (8 h), under light levels of 700 µmol m$^{-2}$ s$^{-1}$, 70% relative humidity to confirm the expected mutations. For this, the youngest leaf of individual plants at 2 weeks post-germination was collected for genotyping. Sample DNA was extracted by using Wizard® Genomic DNA Purification Kit and PCR was performed using primers listed in Supplementary Table 1. The 25 µL PCR reaction was performed according to the manufacturer's instructions and the size-confirmed PCR product was sequenced. Subsequently, the point mutation was confirmed by sequencing using primers listed in Supplementary Table 1. Homozygous plants were backcrossed with wild-type (Cadenza0000) to reduce load of background mutations, followed by self-pollination. For phenotyping, seeds from plants with and without the mutation of interest were put on wet paper enclosed by plastic wrap and kept at 4 °C for 3–4 days, and then transferred to room temperature for phenotyping. Seeds that germinated uniformly were selected and grown in plastic pots containing soil at 24 and 34 °C under 16 h light/8 h dark (100 µmol m$^{-2}$ s$^{-1}$ photosynthetically active radiation, supplied by cool-white, fluorescent tungsten tubes, Osram), and 65–75% air humidity. Phenotyping pictures of 2-week temperature treatment were taken by using Canon digital camera.

**Low red-far/red (R/Fr) treatment**. Seedlings were cultivated on full-strength MS medium without sucrose containing 0.8% agar. Plants were cultivated in white light conditions for 2 days at 22 °C (control) under short-day photoperiod (8 h light/16 h darkness) (Snijders, Microclima 1000, approximately 120 µmol m$^{-2}$ s$^{-1}$ photosynthetic active radiation (PAR)). Plates were subsequently subjected for 6 days to a low R/Fr light ratio (600 nm/730 nm) of 0.09 or remained under control white light conditions with an R/Fr light ratio of 3.04. Low R/Fr ratio was obtained by supplementing the white growth chamber lights with far-red LEDs (Philips

Greenpower LED research module far red), without affecting the photosynthetic active radiation. Spectra were obtained by a Skye SKR100 Display Meter with SKR 110 Red/Far-red sensor. The 8-day-old seedlings were subsequently scanned using a flatbed scanner and hypocotyl lengths were measured using ImageJ image-analysis software (https://imagej.nih.gov/ij/).

**Generation of constructs and transgenic lines**. PCR was performed using Q5® High-Fidelity polymerase according to the manufacturer's instructions (New England Biolabs). The CDS of TOT3 and a fragment of 1999 base pairs upstream of TOT3 CDS (called TOT3 promoter or pTOT3) were amplified by PCR from reverse-transcribed RNA and genomic DNA extracted from Col-0 seedlings, respectively. The CDS of TOI4 and TOI5 was synthesized using the BioXP3200TM system (SGI-DNA) with DNA sequence being modified so that the coding information was retained. For Golden Gate cloning, the TOT3 promoter and TOT3 CDS were cloned into pGGA000 and pGGC000 (ref. [61]). To generate 35S::GFP:TOT3 and pTOT3::GFP:TOT3 constructs, respectively. The respective fragments were assembled into the destination vector pP-GA[61] (Supplementary Table 2). Plant vectors were transformed in *Agrobacterium tumefaciens* C58C1 using the freeze-thaw method[62]. Plant transformation was performed using the floral dip method[63]. All transgenic plants contain a BastaR resistance marker. For site-directed mutagenesis, the pGGC000 (TOT3 CDS) was used as a template for PCR using specific primers containing the mutation site. Then, the DNA template plasmid was digested with DpnI for 1 h before being transformed into DH5α. *E. coli* cells. For the Y2H assay and TAP analysis, the CDS of interest was cloned into the Gateway entry vector pDONR221 and then recombined into pGAL424gate or pGBT9gate. All the primers are listed in Supplementary Table 1.

**Confocal microscopy**. Whole seedlings were immersed in an aqueous solution of 2 or 10 µM *N*-(3-triethylammoniumpropyl)-4-(6-(4-(diethylamino)phenyl)hexatrienyl)pyridinium dibromide (FM4-64). Seedlings were stained for 5 min and then quickly washed with Milli-Q water to remove excessive FM4-64. Seedlings were mounted in Milli-Q between slide and cover glass for imaging with a Zeiss inverted LSM710 confocal laser scanning microscope, equipped with a LD-Plan Neofluar 40x/0.6 Korr M27 or C-Apochromat 40x/1.20 W Korr M27. A 488 nm laser excitation (at 2% power) of a 20 mW argon laser (LASOS) and a spectral detection bandwidth of 493–532 or 493–579 nm was used for detecting eGFP and a 561 nm laser excitation (at 2% power), together with a spectral detection bandwidth of 596–645 or 621–759 nm for detecting FM4-64 signals. Fluorescence signals across the membrane were plotted using ImageJ (https://imagej.nih.gov/ij/). For root-tip imaging, seedlings were infiltrated with water and confocal images were acquired on an inverted Zeiss inverted LSM 710 confocal laser scanning microscope for spectral imaging, using the Zeiss plan apochromat 20x/0.8 M2. The 488 nm laser excitation with a 493–598 nm band-pass emission filter was used for GFP.

**Western blot**. For the TOT3 western blot, finely ground plant materials were homogenized in ice-cold extraction buffer (50 mM Tris-HCl pH 7.5, 150 mM NaCl, 1% NP-40 and a Roche Complete protease inhibitor; 1 tablet per 10 mL). The tubes were centrifuged twice for 20 min at 20,817g at 4 °C. The protein concentration in the supernatant was determined by using the Qubit™ Protein Assay Kit (ThermoFisher, USA) according to the manufacturer's instructions. The proteins were separated on 4–15% SDS-PAGE stain-free protein gel (Bio-Rad Laboratories, Inc., USA), followed by transferring onto a Trans-Blot® Turbo™ Mini PVDF Transfer Packs (Bio-Rad Laboratories, Inc., USA). The membrane for proteins was immune-reacted with anti-HA (Sigma-Aldrich, USA) (1:2000) and mouse IgG HRP linked whole antibody (GE Healthcare, USA) (1:10,000). For the BES1 western blot, 4-day-old seedlings were grown at 28 °C in short-day conditions on solid medium with mock (DMSO) or brassinolide (100 nM). Finely ground plant materials were homogenized in ice-cold homogenization buffer (1% SDS, 25 mM TRIS pH 7.5, 150 mM NaCl, 10 mM DTT and a Roche Complete protease inhibitor 1 tablet per 10 mL). For immunodetection, anti-BES1 antibody (1:5000) was used as primary antibody, anti-Rabbit at (1:10,000) was used as secondary antibody. The proteins were detected by ChemiDoc™ MP Imaging System (Bio-Rad Laboratories, Inc., USA). For the BES1 dephosphorylation assay, the ratio of the dephosphorylated BES1 to the total BES1 protein level was quantified based on the signal intensity. The loading was adjusted to an equal level based on the amount of tubulin. Signal intensities were determined using Image Lab (Bio-Rad).

**Editing of genomic TOI4 and TOI5**. In order to generate loss-of-function *toi4* and *toi5* mutants, we applied the FAST CRISPR-Cas9 system combined with Golden Gate cloning[64]. Two different guide RNAs (gRNAs) were designed for each gene (TOI4 or TOI5) (Supplementary Table 2). The oligo pair with the same gRNA sequence was assembled and ligated into the BbsI-digested entry vector containing the compatible DNA-overhang using T4 DNA ligase (New England Biolabs)[64] (Supplementary Tables 1 and 2). The entry modules were assembled into the destination vector pFASTRK24GW which contains the red fluorescent FAST marker (Supplementary Table 1)[64]. After plant transformation, transgenic T1 seeds that showed a red fluorescent signal were selected. T2 seeds devoid of red fluorescence were selected and T2 seedlings with desired gene editing were retained for

further seeds' propagation and selection of homozygous *toi4* and *toi5* lines. DNA fragments containing the expected mutation sites were amplified for *TOI4* and *TOI5* and the mutations were confirmed by Sanger sequencing. Seedlings which contained homozygous −1 or +1 indel mutations at *TOI4* and *TOI5* loci were selected as *toi4* and *toi5* loss-of-function mutants, since this type of mutation causes a frame shift in the resulting mRNA and subsequently leads to mistranslated proteins. Further, *toi4 toi5* loss-of-function mutants were checked for *TOI4* and *TOI5* expression levels by qPCR using the primers listed in Supplementary Table 1, showing a reduced *TOI4* and *TOI5* mRNA level.

**Yeast two-hybrid assay.** The Y2H assay was performed using the GAL4 system[65]. The prey and bait CDS were recombined into pGAL424gate or pGBT9gate, respectively. The *Saccharomyces cerevisiae* PJ69-4A strain was co-transformed with the bait and prey plasmid using the polyethylene glycol (PEG)/lithium acetate method. Yeast transformants were selected on synthetic defined (SD) media devoid of leucine (Leu) and tryptophan (Trp) (SD-LeuTrp). Three individual colonies were grown overnight in liquid media at 30 °C. The interaction assays were performed by dropping 10-fold dilution of the overnight cultures on SD media lacking Leu, Trp and histidine (His) (SD-LeuTrpHis) as well as SD-LeuTrp as control. Plates were scanned after 3 days incubation at 30 °C.

**RT-qPCR.** Three biological replicates were performed for each temperature condition and each tested gene. RNA was extracted and purified with the RNeasy Mini Kit (Qiagen) according to the manufacturer's instructions for plant RNA extraction. DNA digestion was done on columns with RNase-free DNase I (Promega). The iScript cDNA Synthesis Kit (Biorad) was used for cDNA synthesis from 1 µg of RNA. qRT-PCR was performed on a LightCycler 480 (Roche Diagnostics) in 384-well plates with LightCycler 480 SYBR Green I Master (Roche) according to the manufacturer's instructions. Two housekeeping genes, the *ACTIN-RELATED PROTEIN 7* (*ARP7*) and the *TRANSLATIONAL ELONGATION FACTOR ALPHA* (*EF1α*), were used for normalization of the expression level of the tested genes. All primers are listed in Supplementary Table 1.

**Protein extraction and digestion for phosphoproteomics.** For the *A. thaliana* time course experiment, three independent biological replicates were collected for each time point. finely ground material from 1 g of seedlings was homogenized in 50 mM Tris-HCl buffer (pH 8), 0.1 M KCl, 30% sucrose, 5 mM EDTA, 1 mM DTT and the appropriate amounts of the complete protease inhibitor mixture and the PhosSTOP phosphatase inhibitor mixture (Roche). The samples were sonicated on ice and centrifuged at 4 °C for 15 min at 2500*g* to remove debris. Methanol/chloroform precipitation of proteins was performed on the supernatant[66]. The protein pellets were resuspended in 6 M guanidinium hydrochloride. Alkylation of cysteine residues were performed by adding to a final concentration of 15 mM tris (carboxyethyl)phosphine (TCEP, Pierce) and 30 mM iodoacetamide (Sigma-Aldrich). The sample buffer was exchanged on Illustra NAP columns (GE Healthcare Life Sciences) to 50 mM TEAB buffer (pH 8). One milligram of the proteins was predigested with EndoLysC (Wako Chemicals) at an enzyme-to-substrate ratio of 1:500 (w:w) for 4 h, followed by a digestion with trypsin overnight at an enzyme-to-substrate ratio of 1:100 (w:w). For the comparative phospho-proteome analysis of Col-0 and *tot3-2* seedlings (1 g of plant material) and high-temperature treatment of soybean, four biological replicates were collected for each condition. For these samples, after methanol/chloroform precipitation, proteins pellets were dissolved in 8 M urea. After cysteine alkylation, 3 mg protein were digested with EndoLysC for 4 h. After that, the samples were diluted 8-fold with 1 M urea and proteins were digested further with trypsin overnight. The digest was acidified to pH 3 with trifluoroacetic acid (TFA) and desalted using SampliQ C18 SPE cartridges (Agilent) according to the manufacturer's guidelines, vacuum dried and resuspended in 80% (v/v) acetonitrile, 5% (v/v) TFA.

**Phosphopeptide enrichment.** The re-suspended peptides were incubated with 1 mg MagReSyn® Ti-IMAC microspheres (ReSyn Biosciences) for 20 min at room temperature with continuous mixing. The microspheres were washed once with 60% acetonitrile, 1% TFA, 200 mM NaCl and twice with wash 60% acetonitrile, 1% TFA. The bound phosphopeptides were eluted with three volumes (80 µL) of elution buffer (40% acetonitrile, 5% NH₄OH), immediately followed by acidification to pH 3 using 100% formic acid. The peptides were vacuum dried and stored at −20 °C until LC-MS/MS analysis. The mass spectrometry proteomics data for these experiments have been deposited to the ProteomeXchange Consortium via the PRIDE partner repository with the dataset identifier PXD015468.

**LC-MS/MS analysis.** Samples were analysed via LC-MS/MS on an Ultimate 3000 RSLC nano LC (Thermo Fisher Scientific) in-line connected to a Q Exactive mass spectrometer (Thermo Fisher Scientific). For the Arabidopsis time course experiment, each biological replicate was analysed three times (three technical replicates). The sample mixture was loaded on a trapping column (made in-house, 100 µm internal diameter (I.D.) × 20 mm, 5 µm C18 Reprosil-HD beads, Dr. Maisch, Ammerbuch-Entringen, Germany). After flushing from the trapping column, the sample was loaded on an analytical column (made in-house, 75 µm I.D. × 150 mm, 3 µm C18 Reprosil-HD beads, Dr. Maisch). Peptides were loaded with loading

solvent A (0.1% TFA in water) and separated with a linear gradient from 98% solvent A′ (0.1% formic acid in water) to 55% solvent B′ (0.1% formic acid in water/acetonitrile, 20/80 (v/v)) over 170 min at a flow rate of 300 nL min⁻¹. This was followed by a 5 min wash reaching 99% of solvent B. The mass spectrometer was operated in data-dependent, positive ionization mode, automatically switching between MS and MS/MS acquisition for the 10 most abundant peaks in a given MS spectrum. The source voltage was 3.4 kV and the capillary temperature was set to 275 °C. One MS1 scan (*m/z* 400–2000, AGC target 3 × 106 ions, maximum ion injection time 80 ms) acquired at a resolution of 70,000 (at 200 *m/z*) was followed by up to 10 tandem MS scans (resolution 17,500 at 200 *m/z*) of the most intense ions fulfilling predefined selection criteria (AGC target 5 × 104 ions, maximum ion injection time 60 ms, isolation window 2 Da, fixed first mass 140 *m/z*, spectrum data type: centroid, under fill ratio 2%, intensity threshold 1.7 × 104, exclusion of unassigned, 1, 5–8, > 8 charged precursors, peptide match preferred, exclude isotopes on, dynamic exclusion time 20 s). The HCD collision energy was set to 25% normalized collision energy and the polydimethylcyclosiloxane background ion at 445.120025 Da was used for internal calibration (lock mass).

**Database searching and data analysis.** MS/MS spectra were searched against the TAIR10 database for *A. thaliana* (34,509 entries, version November, 2014) or with The UniProt database (74864 entries, Proteome ID: UP000008827) for *Glycine max* (Soybean) with MaxQuant software (version 1.5.7.4 or version 1.5.4.1, respectively), a program package allowing MS1-based label-free quantification[67,68]. The mass spectrometry proteomics data, the MaxQuant settings and MaxQuant output have been deposited to the ProteomeXchange Consortium via the PRIDE partner repository with the dataset identifier PXD015468. The "Phospho(STY).txt" output file generated by the MaxQuant search was loaded into the Perseus data analysis software (version 1.5.5.3) available in the MaxQuant package[69]. For the time-dependent phosphoprofiling data of Col-0 seedlings transferred to 27 °C, phosphosites that were quantified in at least two out of three biological replicates for at least one time point were retained. Log2 phosphosite intensities were centred by subtracting the median of the entire set of protein ratios per sample. A multiple-sample test (one-way ANOVA) with a *p*-value cut-off of <0.01 was carried out to test for differences between the time points. For the statistical test, the number of randomizations was set at the default value of 250, the technical replicates were preserved during the randomizations. In addition, we retained candidates that were not detected in all biological and technical replicates of at least one time point (and thus could not be subjected to statistical analysis), but for which a phosphosite was detected in one of the other time points. For the soybean phosphoproteome analysis of seedlings upon control and high temperature, phosphosites that were quantified in at least three out of four replicates, from at least one treatment, were retained. Among these phosphosites, the phosphosites not detected in one of the two temperatures were designated as "unique" phosphosites for the other temperature. For the rest of reproducibly quantified phosphosites, log2 phosphosite intensities were centred by subtracting the median of the entire set of protein ratios per sample. A two-sample test (Student's *t*-test) with a *p*-value cut-off of <0.01 was carried out to test for differences between the temperatures. For comparative phosphoproteome profiling of Col-0 and *tot3-2*, phosphosites that were quantified in at least three out of four replicates from at least a genotype were retained (reproducibly quantified phosphosites). Among these phosphosites, the phosphosites not detected in one of the two genotypes were designated as "unique" phosphosites for the other genotype. For the rest of reproducibly quantified phosphosites, log2 phosphosite intensities were centred by subtracting the median of the entire set of protein ratios per sample. A two-sample test with a *p*-value cut-off of <0.01 was carried out to test for differences between the genotypes.

**Identification of Arabidopsis orthologues for wheat/soybean proteins.** For wheat, the orthologous pairs were determined by blasting the *Triticum aestivum* protein database against the non-redundant Arabidopsis protein sequences using BLAST2GO (E-value < 10⁻⁵) software[70]. For simplification, only the Arabidopsis blast hits with the lowest E-value score are reported. Arabidopsis orthologues of soybean proteins were retrieved from the protein annotation database at www.soybean.org (ref. [71]).

**Comparison wheat/soybean/Arabidopsis phosphoproteome.** For the Arabidopsis candidates from the phosphoproteome analysis (Supplementary Data 1), membrane localization was determined using information from SUBA[72]. In addition, we cross-checked the same candidates with a list of available kinases[73,74]. Where available, the Arabidopsis orthologues from the wheat/soybean hits were compared with the Arabidopsis phosphoproteome.

**Tandem affinity purification.** TAP and protein complex data analysis: For TAP analysis, the Gateway system was used to recombine the TOT3 CDS in to pGNGSrhino vector (Invitrogen). The constructs were transformed into *A. thaliana* PSB-D cells[75]. Three pull-downs were performed on NGSrhino-TOT3-expressing transgenic cultures using the cell culture TAP extraction protocol[76], with 1% digitonin added, and a shortened benzonase treatment time of 15 min. To isolate protein complexes, 25 mg total protein extract was incubated for 45 min with 50 mL magnetic IgG antibody bead suspension, prepared in-house as

described[77]. Beads were washed three times with 500 mL TAP extraction buffer and one time with 500 mL TAP extraction buffer without detergent. For on-bead protein digestion, the beads were washed with 500 mL 50 mM $NH_4HCO_3$ (pH 8.0). The wash buffer was removed and 50 mL 50 mM $NH_4HCO_3$ was added together with 1 mg Trypsin/Lys-C and incubated at 37 °C for 4 h. Next, the digest was separated from the beads and overnight incubated with 0.5 mg Trypsin/Lys C at 37 °C. Finally, the digest was centrifuged at 20,800g in an Eppendorf centrifuge for 5 min, the supernatant was transferred to a new 1.5 mL Eppendorf tube, and the peptides were vacuum dried and stored at −20 °C until LC-MS/MS analysis. The peptides were analysed on a Q Exactive (ThermoFisher Scientific)[78]. After MS-based identification of co-purified proteins, specific proteins were detected by comparison against a list of non-specific proteins[79], built from 213 pull-downs with 42 different bait proteins[76]. True interactors that might have been missed because of their presence in the list of non-specific proteins were retained through a semi-quantitative analysis. In this approach, average normalized spectral abundance factors (NSAF) of the identified Arabidopsis proteins in the TOT3 samples were compared against the corresponding average NSAF deduced from the control pull-down dataset. For stringent filtering of specific proteins, only the proteins identified with at least two peptides were retained that were highly (at least 10-fold) and significantly [$-\log_{10}(p\text{-value}(t\text{-test})) \geq 10$] enriched compared to the control dataset. The mass spectrometry proteomics data for these experiments have been deposited to the ProteomeXchange Consortium via the PRIDE partner repository with the dataset identifier PXD015483.

**Co-immunoprecipitation**. Col-0, *pTOT3::GFP:TOT3* seedlings and *35S::GFP* seedlings were grown in similar conditions as used for the time-dependent phosphoproteome profiling. Ten-day-old light-grown seedlings were collected at 0 min at 21 °C and at 60 min after being transferred to 28 °C. For each condition and genotype, three biological replicates were obtained. Next, 1 g of finely ground plant material was mixed thoroughly with 2.2 mL extraction buffer (50 mM TRIS-HCl, 150 mM NaCl, 0.5 mM EDTA, 0.5% NP-40) and sonicated. Debris was removed from the samples by centrifugation, and 2 mL of sample was diluted 5-fold with extraction buffer without NP40, then mixed with 50 μL pre-equilibrated GFP-Trap®_MA beads (ChromoTek) and rotated for 2 h at 4 °C to maximize the protein binding. Subsequently, the solution was removed, the beads were washed three times with wash buffer (50 mM Tris-HCl pH 7.5, 250 mM NaCl), and then once with 1 mL 50 mM TEAB (pH 8.0) (Thermo Fisher). On-bead digestion was performed on the bound proteins in 50 μL TEAB and 0.5 μg trypsin (Promega) for 2 h at 37 °C. The supernatant was retained. The beads were washed twice with 23 μL 50 mM TEAB and the wash solutions were pooled with the previous supernatant. Disulfide bonds were reduced by adding TCEP and iodoacetamide (Thermo Fisher) to a final concentration of 10 mM and 15 mM, respectively, and incubated at 30 °C. The remaining iodoacetamide was quenched by adding DTT to a final concentration of 4 mM. Then 0.5 μg trypsin was added and the sample was incubated overnight at 37 °C to complete the digestion. The digestion was stopped by adjusting the sample to 1% TFA and samples were desalted using C18 Bond Elut tips (Agilent Technologies). MS analysis of the peptides was performed identically to TAP analysis. After MS-based identification of co-immunoprecipitated proteins, proteins were considered as potential interactors of TOT3 if they were present in *pTOT3::GFP:TOT3* samples and not in the other samples; or if the fold-change of the LFQ intensity was 10-fold higher in the *pTOT3::GFP:TOT3* samples, compared to Col-0 and *35S::GFP* samples, either at 21 °C or at 28 °C. The mass spectrometry proteomics data have been deposited to the ProteomeXchange Consortium via the PRIDE partner repository with the dataset identifier PXD015483.

**Recombinant protein purification**. Full-length TOT3 and TOT3$^{D157N}$ and truncated versions of kinase-dead TOI4$^{D188N}$Δ1-38 and TOI5$^{D174N}$Δ1-25 were cloned with an N-terminal MBP tag and a C-terminal 6xHis tag into the pET vector backbone using NEBuilder HiFi DNA Assembly kit according to the manufacturer's instruction. BL21(DE3) competent cells were used for recombinant production of the proteins. Proteins were extracted in lysis buffer containing 20 mM TRIS pH 7.5, 150 mM NaCl, 1 mM TCEP and 10% glycerol. The proteins were purified using HisPur$^{TM}$ Ni-NTA resin (Thermo Scientific) and eluted in the lysis buffer containing 250 mM imidazole.

**In vitro kinase assay and mass spectrometry analysis**. Kinase assays were performed in the reaction buffer containing 20 mM Tris–HCl, pH 7.5, 5 mM EGTA, 1 mM DTT, 50 μM ATP and 10 mM MgCl2 at 28 °C for 1 h. A reaction using kinase-dead TOT3$^{D157N}$ was performed as a negative control. The samples were separated by SDS-PAGE. After being stained with Coomassie Blue, protein bands were excised and processed for MS analysis[80]. MaxQuant (v. 1.6.11.0) was used to search for phosphopeptides in the samples, using a search database combining the BL21(DE3) proteome and the sequences of MBP- and His-tagged TOT3, TOI4$^{D188N}$Δ1-38 and TOI5$^{D174N}$Δ1-25.

**Transient expression in tobacco and mass spectrometry analysis**. The plasmids pB7WGF2::GFP-TOT3, pB7WGF2::GFP-TOT3$^{D157N}$; pB7WGR2::RFP-TOI4$^{D188N}$ and pB7WGR2::RFP-TOI5$^{D174N}$ were generated using the Gateway$^{TM}$ system and used to transformed the *A. tumefaciens* C58. GFP-tagged TOT3/

TOT3$^{D157N}$ were co-infiltrated with RFP-TOI4$^{D188N}$ or RFP-TOI5$^{D174N}$ in 6-week-old of tobacco leaves. The samples were harvested after 3 days in four biological replicates. The finely ground material was suspended in homogenization extraction buffer (50 mM Tris-HCl pH 8.0, 150 mM NaCl 0.5% and 0.5% NP40) containing appropriate amounts of the cOmplete™ protease inhibitor mixture (Roche) and the PhosSTOP phosphatase inhibitor mixture (Roche). The samples were incubated by rotation at 4 °C for 30 min to maximize extraction of proteins. Cell debris was removed from the supernatant by centrifugation, and 10 mg total protein from each sample was used for co-immunoprecipitation, then 50 μL equilibrated RFP Trap (ChromoTek) magnetic beads per sample was added into 10 mg total protein of each sample and incubated by rotation for 1 h at 4 °C. Beads were washed three times with wash buffer (50 mM Tris-HCl pH 7.5, 250 mM NaCl), followed by washing once with 1 mL 50 mM TEAB (pH 8.0) (Thermo Fischer). On-bead digestion was performed on the bound proteins with 0.5 μg trypsin (Promega) in 50 μL 50 mM TEAB (pH 8.0) for 2 h at 37 °C. The supernatants were retained, beads were washed twice and the wash fractions were pooled with the supernatants. Samples were treated with 10 mM TCEP and 15 mM iodoacetamide (Thermo Fisher) for 30 min at 30 °C to reduce and carbamido-methylate cysteine residues, then 0.5 μg trypsin was added to the samples for further digestion and these were incubated overnight at 37 °C. The digestion was stopped by adjusting the sample to 1% TFA and samples were desalted using C18 Bond Elut tips (Agilent Technologies) and subjected to LC-MS/MS analysis as described above. MS/MS spectra were searched against the combined protein database of GFP/RFP, GFP-TOT3/ TOT3$^{D157N}$ and RFP-TOI4$^{D188N}$/TOI5$^{D174N}$, and using the *Nicotiana benthamiana* proteome downloaded from SolGenomics database containing 57,140 protein entries by the MaxQuant software (version 1.6.10.43) using UseGalaxy.be server. Data processing was similar to the analysis of Arabidopsis phosphoproteome. All the phosphosites with at least 3 valid values at least in one treatment group were retained as reproducibly quantified phosphosites for statistical analysis. The two-sample test with $p < 0.05$ was carried out to test the differences among the treatments.

**Statistics and reproducibility**. For box plots, the lower Tukey-based whisker shows the smallest value that is greater than the lower quartile minus 1.5 × interquartile range, the upper Tukey-based whisker shows the greatest value that is smaller than the upper quartile plus 1.5 × interquartile range and data points outside this range (outliers) are plotted as individual dots.

**Venn diagrams and heatmaps**. Comparing lists was performed using Venny v2.1 (https://bioinfogp.cnb.csic.es/tools/venny/). Heatmaps were generated using Perseus v1.5.5.3 (ref. [69]) or MeV v4.9.0 (ref. [81]).

**Reporting summary**. Further information on research design is available in the Nature Research Reporting Summary linked to this article.

## Data availability
The raw mass spectrometry proteomics datasets are publicly available through ProteomeXchange with identifiers PXD015468 and PXD015483. All other data are available from the corresponding author upon request. Source data are provided with this paper.

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

## Acknowledgements
We thank Barbara Wrzesinska for technical help and Bert De Rybel for providing critical comments on the manuscript. We thank the following colleagues for providing materials: Pablo D. Cerdán (*phyb-9*), Salomé Prat (*pif4-101*) and Yanhai Yin (anti-BES1 antibody). We also express our gratitude to Thomas Jacobs for advice on generating CRISPR mutants for *TOI4* and *TOI5* and Martha Ramirez for soybean growth. The EMS-mutagenized population of bread wheat cv. Cadenza was developed by Andy Phillips at Rothamsted. We acknowledge the Germplasm Resources Unit (GRU) at JIC for providing wheat seeds. L.D.V was a recipient of the VIB International PhD Scholarship in Life Sciences. X.X., T.Z and Y.W. are supported by grants from the Chinese Scholarship Council. A.L. was supported by a grant from the North Carolina Soybean Producers Association.

## Author contributions
I.D.S and K.G. initiated and managed the project and designed experiments. L.D.V., E.S., D.d.J., X.X, M.v.Z., T.V., T.Z., A.L., L.P., Y.W. N.D.W. and B.v.d.C. performed experiments. I.D.S., K.G., M.v.Z., C.U., G.D.J., E.R. and D.V.D. contributed ideas, interpreted results and critically revised the manuscript. All authors discussed the results and approved the manuscript.

## Competing interests
The authors declare no competing interests.
