## [Peer Review File · Nature Communications]

REVIEWER COMMENTS

Reviewer #1 (Remarks to the Author):

The manuscript by Vu et al discusses a role for a protein "Target of Temperature 3", which appears to be differentially phosphorylated in response to temperature. A screen for changes in phosphorylation in proteins by phosphoproteomics in response to a shift to 27°C identified 212 diff phos sites in 180 proteins. (additionally, 12 sites added) – They do a similar work in soybean/wheat and Arabidopsis and identified 42 TOTs (Targets of Temperature). Of these TOTs looked for membrane-associated kinases, and found TOT3 to be the only candidate. This paper then goes on to characterise the role of TOT3 in thermal response. The authors claim that the "warm temperature-mediated growth in flowering plants is controlled by a conserved signalling complex".

This paper is interesting and it has a different perspective starting from the protein components and differential phosphorylation etc. The authors managed to find a protein that seem to be of interest in the context of thermal responses. The experiment and the data presented are quite good. Although I am not an expert on proteomics, the data they have presented appears to be pretty nice. I only have some minor comments on the data part of this paper.

The major problem with this paper is that the authors claim too much, which I do not believe is supported by the data they presented. Too many loose ends are there and the authors appear to be quite comfortable to reach conclusions without giving considerations for alternative explanations. There are several instances like this, which I will go a bit more in detail below.

Lines 115 - Picloram experiments: I do not really believe their picloram experiment tells anything about the involvement or non-involvement of TOT3 in general growth. The authors are using comparative datasets in different ways. For example, in line 115, the authors state "tot3-2 still responded to these cues". This somehow hints that tot3-2 is not responsive to warm-temperature cues, which is not true (Fig 2b). Tot3-2 mutants are still responsive to warm-temperature cue as well (Fig 2b). Fig 2b and 2e pretty much shows the same effect (which is expected, given picloram is an auxin analogue). I would prefer to see whether a G x E (or drug) effect is significant and how much that explains in each of these conditions, which will provide a quantitative comparison of the different cues.

PHYB/PIF genetics: From the data it is very difficult to draw the conclusion that TOT3 is independent of phyB and PIF4 signalling. The double mutants of phyB and tot3 do support the interpretation of the additive effect. I do not believe the pif4 data can be interpreted as additive effect. Each individual mutant effect is more than 50%, which will mean that the double mutants will saturate. So, I do not believe they are in a position to draw any conclusion from the genetic analysis with pif4/tot3 doubles. Thus they cannot conclude that they act independently.

I am not convinced with their data and arguments presented in Fig 3/Extended Data 11 etc. In my view their analysis lacks the statistical rigour. Also, some of the logic that treating plants with auxin will restore to the same extent as the genetic complementation does not make sense. While this can happen, it is not a given, particularly if there are multiple pathways involved. However, the authors appear to be quite comfortable to reach strong conclusions.

158-160: "Reduction in TOI4 and TOI5 peptides in tot3-2 at 28°C compared to Col" does not really tell much about their role in thermal signalling. If their interaction is temperature-independent, it is also possible that the stability of complex may be higher than individual proteins that one might observe differences.

Lines 170-171: It is not clear to me what data shows TOT3 signalling requires TOI4 and TOI5.

Lines 185-187: What is TOT3 impact is independent of BRI/BZR? Would you not get the same results. If TOT3 acts upstream in a cascade then there may be no role/requirement for BRI/BZR and one would get similar results.

Additional minor comments:

1) Fig. 1

Not really sure whether Fig 1f & g need to be in the main figure. The label for 21°C is unreadable

2) Fig. 3

I think here as well the authors should directly compare expression levels in a GXE analysis through ANOVA. It is not clear to me how the t-test was applied.. Is it on single fold change value...Please explain. From the looks it appears to me some of them are significantly impacted (e.g., HSP70, IAA29 for instance).

3) Extended Fig 12d.

Please add the control transgenic complementing line for comparison and show the statistical differences with that.

4) Fig 4F.

Their representative pictures show that the *toi4 toi5* double mutants may not yet have opened their cotyledons (like Col or *tot3*) and may still be growing.. Have the authors considered that? Or is it just a picture that may not quite be representative of the mutants.

5) Fig4h.

Where is the Col control with BL?

Reviewer #2 (Remarks to the Author):

In this paper, the authors conducted phosphoproteome analyses of Arabidopsis seedlings and soybean leaves to understand signaling mechanisms that regulate thermomorphogenesis. Comparative analysis also by utilizing published wheat phosphoproteomics data set identified TOT (Targets Of Temperature) proteins that are commonly phospho-regulated in at least two of analyzed plant species upon warm temperature treatments. Among 42 TOTs, the authors focused on the single potentially membrane located protein kinase, which is TOT3 that belongs to MAP4K family. The authors have shown that Arabidopsis *tot3* T-DNA insertion mutants have defect in thermomorphogenesis. Complementation analyses suggested that kinase activity of TOT3 is required for the observed thermomorphogenesis response. Genetic analyses indicated that TOT3 acts independently of known PIF4 and phyB thermosignaling pathways. Furthermore, the authors conducted interactome analyses and showed that TOT3 interacts with other MAP4Ks, TOI4 and TOI5. Arabidopsis *toi4 toi5* double mutant displayed hypocotyl growth defect only in darkness, which may suggest that TOI4 and TOI5 are partly required for TOT3-dependent signaling.

This work will definitely shed new light on our understanding of molecular mechanisms underlying thermomorphogenesis. Overall experiments are well-designed and I very much enjoyed reading the manuscript. However, I felt that some major conclusions drawn in this manuscript and obviously title of the manuscript are not fully supported by the provided data, which can be rather misleading, and therefore it need to be properly adjusted by performing additional experiments or intensive rewriting.

I guess the manuscript was transferred from other journal, and I do not know if the Nature Communications accept the current format in general, but I strongly suggest to have introduction and discussion sections for easier reading. For instance, information about plant MAP4Ks need to be provided, because not many people are familiar with MAP4K homolog in plants. There are several studies described on functions of plant MAP4Ks, and recently MAP4K4/TOT3 was shown to regulate plant immunity. Obviously, more details about thermomorphogenesis are better to be introduced and discussed.

Phenotype of wheat *tot3* mutant lines is not very much clear from the provided data. The authors should apply statistical analysis to draw a solid conclusion. What is known about thermomorphogenesis in monocots, and which phenotypes are expected? Anyway, in its current

form I cannot agree that TOT3 homolog/ortholog(?) also regulates thermomorphogenesis in wheat/monocots and the statement that TOT3-mediated signaling mechanism is evolutionary conserved. Based on the provided supplementary tables and published data, it seems that TOT homolog was only found to be phospho-regulated in wheat spikelets but not in wheat and soybean leaves. Do these data fit to the stated/observed phenotype of the wheat tot3 mutants?

The authors should perform phylogenetic analysis to provide phylogenetic tree of plant MAP4K to show relationship between Arabidopsis TOT3 and wheat TOT3, which also tell us how TOT3 or other MAP4Ks are conserved among plant species to discuss evolutionary context. Related, conservation of the identified TOT3 phospho-sites should also be analyzed/provided. In addition, I suggest to provide phosphorylation kinetics of Arabidopsis TOT3 as a graph.

There is no single clear evidence that TOI4 and TOI5 function together with TOT3 in thermomorphogenesis. The phosphoproteomics data (Fig.4C) may naively suggest that TOI4 and TOI5 are targeted by TOT3 during the response. However, the authors should keep in mind that there are possibilities that mRNA or protein levels of TOI4 and TOI5 are affected in the tot3-2 mutant, which data are not provided, and thereby the reduced phosphorylation levels of TOI4 and TOI5 in the tot3-2 mutant may not be due to defect in TOT3 kinase dependent signaling. Moreover, there are no evidence that homologs/orthologs of TOI4 and TOI5 have similar function in other plant species. Therefore, I would say that 'a conserved signalling complex' is an overstatement. The authors may evaluate mRNA and protein levels, and perform in vitro kinase assay to show that TOT3 does phosphorylate TOI4 and TOI5 at the identified phospho-sites.

Other comments need to be addressed

1. Please explain and describe at the method section how orthologous pairs among Arabidopsis, wheat and soybean were defined. I think that there are many cases that relationships are not one-to-one, and it is important to know how these are handled.
2. Please provide a list/table of TOTs with gene IDs of Arabidopsis, wheat and soybean.
3. It seems that most of TOTs are based on Arabidopsis and soybean data, which means that many of TOTs could function only in dicots. Above mentioned list would make the point clearer, and the authors should describe/discuss more in detail about these.
4. Please provide a volcano plot for co-IP data.
5. Please provide details about LC-MS setting and methods used for measurements.
6. It was not clear to me how Arabidopsis phosphoproteome data was (statistically) analyzed. Which 'multiple sample test' was used? How missing values were treated? How biological and technical replicates were treated for analysis? Please explain more in detail.

Optional

Although it's published data, it would be helpful for readers to provide reformatted wheat phosphoproteomics data used in this study as a supplementary table.

We have addressed all comments and suggestions (see below), and we have highlighted all new sentences/figures in yellow throughout the main manuscript and supplementary information.

Reviewer #1 (Remarks to the Author):

The major problem with this paper is that the authors claim too much, which I do not believe is supported by the data they presented. Too many loose ends are there and the authors appear to be quite comfortable to reach conclusions without giving considerations for alternative explanations. There are several instances like this, which I will go a bit more in detail below.

REPLY: As suggested, we have extensively rewritten the manuscript to remove non-supported claims, such as the link with the regulation of general growth, and we have added, where relevant, alternative explanations. All new sentences/paragraphs and (Supplementary) Figure panels are highlighted in yellow.

Lines 115 - Picloram experiments: I do not really believe their picloram experiment tells anything about the involvement or non-involvement of TOT3 in general growth. The authors are using comparative datasets in different ways. For example, in line 115, the authors state “tot3-2 still responded to these cues”. This somehow hints that tot3-2 is not responsive to warm-temperature cues, which is not true (Fig 2b). Tot3-2 mutants are still responsive to warm-temperature cue as well (Fig 2b). Fig 2b and 2e pretty much shows the same effect (which is expected, given picloram is an auxin analogue). I would prefer to see whether a G x E (or drug) effect is significant and how much that explains in each of these conditions, which will provide a quantitative comparison of the different cues.

REPLY: The picloram experiment has now been re-interpreted to only emphasize the similar response of both Col-0 and tot3-2 to the auxin analogue (lines 196-200). Two-way ANOVA is now used throughout the manuscript when applicable to show the G x E effect. We have toned down or re-interpreted the responses of tot3 to different cues, and further supported a PIF4-independent role with additional data (also see next comment; lines 201-218 and Figure 4e-h).

PHYB/PIF genetics: From the data it is very difficult to draw the conclusion that TOT3 is independent of phyB and PIF4 signalling. The double mutants of phyB and tot3 do support the interpretation of the additive effect. I do not believe the pif4 data can be interpreted as additive effect. Each individual mutant effect is more than 50%, which will mean that the double mutants will saturate. So, I do not believe they are in a position to draw any conclusion from the genetic analysis with pif4/tot3 doubles. Thus they cannot conclude that they act independently.

REPLY: We have included new data for tot3-2, pif4-101, and the tot3-2 pif4-101 double mutant grown at 21 and 28 °C in darkness (Figure 4). This set-up allows us to look into temperature-induced hypocotyl elongation independently from the photoperiod which

also controls the activity of PIF4. In this set-up, the response of *pif4-101* to high temperature is similar to Col-0. However, *tot3-2* and *tot3-2 pif4-101* hypocotyl length is largely not responsive to the elevated temperature. This indicates that there is a mechanism controlling temperature-mediated hypocotyl elongation via TOT3 and independently from the photoperiod and PIF4. This observation is now described in the text (lines 201-218). Furthermore, qPCR data also shows that the expression of PIF4 and its target genes are similar in both Col-0 and *tot3-2* mutant (Figure 4e,h). We agree that the double mutants alone do not support the interpretation of an additive effect, but with the (new) genetic and the qPCR results showing that PIF4-mediated gene expression is not affected in *tot3*, we are convinced that we can conclude that they are part of independent pathways.

I am not convinced with their data and arguments presented in Fig 3/Extended Data 11 etc. In my view their analysis lacks the statistical rigour. Also, some of the logic that treating plants with auxin will restore to the same extent as the genetic complementation does not make sense. While this can happen, it is not a given, particularly if there are multiple pathways involved. However, the authors appear to be quite comfortable to reach strong conclusions.

REPLY: We have added data for 21 °C throughout the manuscript and now use two-way ANOVA (genotype x treatment) with Tukey post-hoc testing to strengthen the statistics of our analysis. Following the remark of the reviewer, we have removed the picloram treatment at high temperature and we have rephrased our conclusion to emphasize that the effect of the picloram treatment is similar in both Col-0 and *tot3-2* mutant at 21 °C (Supplementary Figure 15b), indicating that auxin response is not affected in the mutant (which is also reflected in the largely unaltered expression of auxin-responsive genes in *tot3* compared to Col-0) (Figure 4e and Supplementary Figure 15a).

158-160: "Reduction in TOI4 and TOI5 peptides in tot3-2 at 28°C compared to Col" does not really tell much about their role in thermal signalling. If their interaction is temperature-independent, it is also possible that the stability of complex may be higher than individual proteins that one might observe differences.

REPLY: We agree that the phosphopeptide level alone is not sufficient to make this claim. We therefore added genetic data on hypocotyl length of *toi4* and *toi5* mutants at high temperature, also in combination with *tot3* (lines 260-267 and Figure 5d-e). The result indicated a redundant role between TOT3 and TOI4. In addition, we confirmed that TOI5 is phosphorylated downstream of TOT3 (although this is likely indirect). We have – in addition – added alternative explanations as suggested (lines 241-244).

Lines 170-171: It is not clear to me what data shows TOT3 signalling requires TOI4 and TOI5.

REPLY: We agree that we had phrased this in a confusing way. We now included new data comparing the *tot3 toi4*, *tot3 toi5* and *toi4 toi5* double mutants (lines 260-267 and Figure 5d-e). In SD condition, the *tot3 toi4* response to high temperature is mostly

abolished, while *tot3 toi5* showed only a slightly reduced response compared to the *tot3* single mutant. Interestingly, the *toi4* and *toi5* single mutants and the *toi4 toi5* double mutant respond similarly to wild-type plants, suggesting that additional redundant MAP4Ks are involved. Alternatively, the genetic interaction indicates that TOT3 is crucial in the pathway, with TOI4 playing an important role, whereas TOI5 only plays a minor role. We have added the different possible explanations in the manuscript (lines 347-354).

Lines 185-187: What is TOT3 impact is independent of BRI/BZR? Would you not get the same results. If TOT3 acts upstream in a cascade then there may be no role/requirement for BRI/BZR and one would get similar results.

REPLY: We believe that TOT3 does not work independently of BRI1/BZR1. First, we observed phosphorylation changes in BR signalling-associated proteins in *tot3* versus wildtype (Supplementary Figure 21b). Second, our data imply that brassinosteroid signalling might be severely impaired in the *tot3 toi4 toi5* mutant because this higher order mutant strongly resembles a *bri1* mutant (Supplementary Figure S24), suggesting a role in BR signaling. Third, we now also included data of brassinazole (BRZ) treatment for *bzr1-1D* and *tot3-2 bzr1-1D* mutants. The BRZ-insensitive phenotype of *bzr1-1D* plants is highly reduced in the *tot3-2 bzr1-1D* double mutant (Figure 6c-d). Finally, expression analyses of several BZR1 target genes also support a role for TOT3 in regulating BZR1 activity (Figure 6e-f and Supplementary Figure 23). Taken together, the data showed that TOT3 does indeed affect BR signalling and BZR1 activity (lines 282-285/290-311).

Additional minor comments:

1) Fig. 1 - Not really sure whether Fig 1f & g need to be in the main figure. The label for 21°C is unreadable

REPLY: We have moved Col-0 data to Supplementary Figure 5. The labels have been improved for visibility.

2) Fig. 3 - I think here as well the authors should directly compare expression levels in a GXE analysis through ANOVA. It is not clear to me how the t-test was applied.. Is it on single fold change value...Please explain. From the looks it appears to me some of them are significantly impacted (e.g., HSP70, IAA29 for instance).

REPLY: We have now included the 21 °C data for all analyses and applied a two-way ANOVA with Tukey post-hoc test. The fold-changes between the temperatures are also indicated when there is a significant difference.

3) Extended Fig 12d - Please add the control transgenic complementing line for comparison and show the statistical differences with that.

REPLY: As suggested, we have repeated the experiment with the transgenic TOT3 complementing line and performed new statistical analyses (Supplementary Figure 14b).

4) Fig 4F - Their representative pictures show that the *toi4 toi5* double mutants may not yet have opened their cotyledons (like *Col* or *tot3*) and may still be growing.. Have the authors considered that? Or is it just a picture that may not quite be representative of the mutants.

REPLY: We looked into this and selected a better representative picture (Figure 5d).

5) Fig4h - Where is the *Col* control with BL?

REPLY: We have repeated the experiment and the Figure now includes the *Col* + BL treatment (Figure 6b).

Reviewer #2 (Remarks to the Author):

However, I felt that some major conclusions drawn in this manuscript and obviously title of the manuscript are not fully supported by the provided data, which can be rather misleading, and therefore it need to be properly adjusted by performing additional experiments or intensive rewriting.

REPLY: Since our previous claims were perceived as too broad by the reviewers, we have rewritten our manuscript as suggested with a more defined focus on the light-independent activity of TOT3 and we have added several additional experiments to support our claims. All new sentences/paragraphs and (Supplementary) Figure panels are highlighted in yellow.

I guess the manuscript was transferred from other journal, and I do not know if the Nature Communications accept the current format in general, but I strongly suggest to have introduction and discussion sections for easier reading. For instance, information about plant MAP4Ks need to be provided, because not many people are familiar with MAP4K homolog in plants. There are several studies described on functions of plant MAP4Ks, and recently MAP4K4/TOT3 was shown to regulate plant immunity. Obviously, more details about thermomorphogenesis are better to be introduced and discussed.

REPLY: We have rewritten the manuscript, including a more extensive introduction (lines 64-85), and added the suggested elements (lines 130-133) in the revised version.

*Phenotype of wheat *tot3* mutant lines is not very much clear from the provided data. The authors should apply statistical analysis to draw a solid conclusion. What is known about thermomorphogenesis in monocots, and which phenotypes are expected? Anyway, in its current form I cannot agree that TOT3 homolog/ortholog(?) also regulates thermomorphogenesis in wheat/monocots and the statement that TOT3-mediated signaling mechanism is evolutionary conserved. Based on the provided supplementary tables and*

published data, it seems that TOT homolog was only found to be phospho-regulated in wheat spikelets but not in wheat and soybean leaves. Do these data fit to the stated/observed phenotype of the wheat tot3 mutants?

REPLY: In the revised manuscript, we now elaborated on thermomorphogenesis in monocots and pinpointed a phenotype for our studies. To address the reviewer's issue, we repeated the phenotyping of wheat *tot3* mutants, also in view of our unpublished data that diverse wheat varieties display a distinct growth profile across a temperature gradient, each with its own growth-promoting temperature. In the revised version, we now describe differential phenotypes of wheat TILLING lines defective in different TOT3 homeologs relative to wild type (lines 151-165), including the appropriate statistical analyses. We included two distinct temperature ranges, the growth-promoting (14-24°C) and growth-repressing (24-34°C) temperature range. For the growth promotion, we focused on the leaf sheath, and showed that the visible leaf sheath is significantly shorter in wheat *tot3* mutants at 24°C (Figure 3 and Supplementary Figure 11). This measurement is a proxy for the growth rate and the emergence of new leaves. For the growth repression we looked at overall seedling size. At growth-repressing temperature, wheat *tot3* mutants are significantly shorter (Supplementary Figure 12).

While we indeed only found differential phosphorylation of TOT3 orthologs in wheat spikelets, we also found TOI4 and TOI5 orthologs in the soybean and wheat leaf data. This suggests that components of the TOT3 signalosome are conserved and may also be involved in high temperature responses in these species. However, given that mass spectrometry does not give a full proteome-wide overview in every independent analysis, the lack of identifying TOT3 phosphorylation does not necessarily indicate that it does not occur in wheat leaves. Because of these limitations (which we added to the manuscript: lines 375-380), we opted to validate our hypothesis by genetic means and for this we focused on *tot3* alleles in wheat. This indeed supported a role for TOT3 in high temperature-mediated growth in wheat.

The authors should perform phylogenetic analysis to provide phylogenetic tree of plant MAP4K to show relationship between Arabidopsis TOT3 and wheat TOT3, which also tell us how TOT3 or other MAP4Ks are conserved among plant species to discuss evolutionary context. Related, conservation of the identified TOT3 phospho-sites should also be analyzed/provided. In addition, I suggest to provide phosphorylation kinetics of Arabidopsis TOT3 as a graph.

REPLY: This is a good suggestion. In fact, we have been working on an in-depth phylogenetic analysis of the MAP4K family. This manuscript is being considered for publication by *Frontiers in Plant Science* and we therefore do not choose to include it in the present manuscript. We however added a dedicated comparative phylogenetic tree to illustrate the relationship between Arabidopsis and wheat MAP4Ks (Supplementary Figure 10a). Herein, wheat TOT3, TOI4 and TOI5 orthologs are clearly visible. The conservation of identified TOT3 phosphosites is now addressed in the discussion (Supplementary Figure 26). For the phosphorylation profiles of wheat TOT3, TOI4 and

TOI5, we now present all the possible homeologues in the graph with the phosphopeptide sequence as reference (Supplementary Figures 10b and 17). To avoid confusion, we only use the first isoform of all wheat protein IDs to indicate the position of the phosphosite in the protein sequences. We would like to remark that the position of the phosphosites may vary from isoform to isoform, so the position number can be slightly different from the previous version of the manuscript, but the phosphosite and the phosphopeptide are identical. It should be noted that all the wheat protein IDs have been modified as follows, the 11th digit has been changed from 1 to 2, e.g TraesCS1B01G199100 is changed to TraesCS1B02G199100, in order to make the IDs searchable in the current updated public database (ENSEMBL, IWGSC).

For the second comment, we assume the reviewer refers to TOT3 phosphorylation in the original data set in Arabidopsis? We have now added this as the supplemental Figure 4a.

There is no single clear evidence that TOI4 and TOI5 function together with TOT3 in thermomorphogenesis. The phosphoproteomics data (Fig.4C) may naively suggest that TOI4 and TOI5 are targeted by TOT3 during the response. However, the authors should keep in mind that there are possibilities that mRNA or protein levels of TOI4 and TOI5 are affected in the tot3-2 mutant, which data are not provided, and thereby the reduced phosphorylation levels of TOI4 and TOI5 in the tot3-2 mutant may not be due to defect in TOT3 kinase dependent signaling. Moreover, there are no evidence that homologs/orthologs of TOI4 and TOI5 have similar function in other plant species. Therefore, I would say that ‘a conserved signalling complex’ is an overstatement. The authors may evaluate mRNA and protein levels, and perform in vitro kinase assay to show that TOT3 does phosphorylate TOI4 and TOI5 at the identified phospho-sites.

REPLY: We now included new data comparing the double mutants of *tot3 toi4*, *tot3 toi5* and *toi4 toi5* with wild type and the *tot3* single mutant (lines 259-267 and Figure 5d-e). In SD condition, *tot3 toi4* response to high temperature is mostly abolished, while *tot3 toi5* showed a slightly reduced response compared to the *tot3* single mutant. Interestingly, the *toi4* and *toi5* single mutants and the *toi4 toi5* double mutant responds similarly as wild-type plants. The genetic interaction indicates that TOT3 is crucial in the pathway, with TOI4 playing an important role, whereas TOI5 only plays a minor role. In addition, we added the expression level of TOI4/5 in *tot3-2* (Supplementary Figure 16b) as suggested, and this showed no difference between Col-0 and *tot3-2*. Furthermore, in our *in vitro* kinase assays, the detection of phosphorylation of wild-type TOT3 but not kinase-dead TOT3 indicates that wild-type TOT3 is indeed active *in vitro* (lines 174-177 and Supplementary Figure 14b). However, we observed no phosphorylation of TOI4 and TOI5. This indicates that TOT3 may not phosphorylate TOI4 and TOI5 directly and changes in TOI4 and TOI5 phosphorylation level *in planta* result from an unknown (and likely indirect) mechanism. Nevertheless, we strengthened the observation that at least TOI5 is downstream of TOT3 through transient expression in tobacco and analyses of its phosphorylation status (lines 245-258 and Supplementary Figure 18). In this experiment, we indeed see changes in phosphorylation that cannot be explained by changes in protein

level alone. In addition, the change in TOI4 levels in tobacco is opposite to the change in the *tot3* mutant. We have toned down the conclusion on the conserved signalling mechanism (indeed, we have not followed up on TOI4 and TOI5 in other species), and provided alternative explanations as requested (lines 334-362).

Other comments need to be addressed

1. Please explain and describe at the method section how orthologous pairs among Arabidopsis, wheat and soybean were defined. I think that there are many cases that relationships are not one-to-one, and it is important to know how these are handled.

REPLY: This is a good suggestion and has now been added to the Method section (lines 857-862). In short, for the wheat data, the orthologous pairs were determined by blasting the wheat protein database against the non-redundant Arabidopsis protein sequences using BLAST2GO (E-Value < 10^{-5}) software as described previously (Vu et al., 2018, *J Exp Bot*). For simplification, we only reported the Arabidopsis blast hits with the lowest E-value score in the Tables. We agree that in many cases those relationships are not one to one, especially since wheat has multiple copies of the same gene in its genome. The duplicated Arabidopsis orthologues are now highlighted in Table S3. The relationship between wheat and Arabidopsis MAP4Ks which are the focus of our manuscript is now illustrated in the phylogenetic tree (Supplementary Figure 10a). The details of the phylogenetic relationships between plant MAP4Ks are described in another manuscript (Pan et al, 2021, *Front Plant Sci*, submitted). For soybean, Arabidopsis orthologs were retrieved from genome and protein annotation database at <https://www.soybase.org/>

2. Please provide a list/table of TOTs with gene IDs of Arabidopsis, wheat and soybean.

REPLY: Thank you for this suggestion. This information has been now added to Table S3. We also added the E-value from the database blast in the table to illustrate the confidence of the orthologue pairs.

3. It seems that most of TOTs are based on Arabidopsis and soybean data, which means that many of TOTs could function only in dicots. Above mentioned list would make the point clearer, and the authors should describe/discuss more in detail about these.

REPLY: We trust that the above-mentioned list makes this clearer, but at this stage we would prefer not to make (functional) statements on dicots versus monocots. We are currently performing more in-depth analyses in soybean, wheat and *Marchantia* to address such a hypothesis, but we think that this is outside the scope of this manuscript. It should be noted that the list of TOTs presented here will be extended and refined when additional phosphoproteome data become available.

4. Please provide a volcano plot for co-IP data.

REPLY: We prefer to be very stringent in our selection, and therefore we did not include interacting proteins that were only slightly, but significantly, more present compared to the control, but rather included only those proteins that were exclusively present in our pull-down assays (or not present compared to a control library). To generate volcano plots however, the missing values have to be imputed with a value $\neq 0$. If we replace these missing values from the normal distribution of the samples (DOI: 10.1038/nmeth.3901) and apply Student's t-test ($p < 0.01$, $FC \geq 5$) (see below), we lose the clean qualitative 'absence/presence' criterium. While we do retain TOI4 and TOI5 as interactors, we noticed that a few previously identified interactors (that were clearly different in our stringent approach) were not significantly different anymore between control and GFP::TOT3 samples after the imputation of the missing values, which computationally adds non-real values (e.g. PDI9 – see below).

We therefore opted to not analyze the data using the volcano plots in order not to lose such interactors. Regardless of this decision, the analysis will not affect the manuscript since both TOI4 and TOI5 could be identified as interactors in both approaches. However, if the reviewer feels strongly about this, we can include both analyses in the manuscript if desired.

5. Please provide details about LC-MS setting and methods used for measurements.

REPLY: We have added these details in the methods (Supplementary Information).

6. It was not clear to me how Arabidopsis phosphoproteome data was (statistically) analyzed. Which 'multiple sample test' was used? How missing values were treated? How biological and technical replicates were treated for analysis? Please explain more in detail.

REPLY: In all of our analyses, we opted not to replace the missing values, since this may lead to aberrant statistics (an example was given above for the IP data above), and this is also the case for the Arabidopsis phosphoproteome data. The technical replicates were defined in Perseus as a separate subgroup of the biological replicates. For the statistical analysis, we used one-way ANOVA for the multiple sample test. The number of randomizations used to generate null distribution to calculate permutation-based FDR is set at the default value of 250. Here, the technical replicates were preserved during the

randomizations (since we do not compare the technical replicates within a time point with each other).

Optional - Although it's published data, it would be helpful for readers to provide reformatted wheat phosphoproteomics data used in this study as a supplementary table.

REPLY: We agree with this suggestion and have added this as Supplementary Data S4.

REVIEWER COMMENTS

Reviewer #1 (Remarks to the Author):

Apologies for the delay.

Authors have addressed most of my concerns in a reasonable manner and have toned down some of their claims appropriately. I do not have any major concerns. I have 3 minor concerns that the authors could address.

1) There are differences with previously published data on inhibiting the BRZ signaling (with ref 41). Authors report a growth promotive effect at lower temperatures and growth inhibitory effect at higher temperatures. While the authors state that "in their hands" this is what they see, both can not be true. It would be better if the authors confirm and provide a potential explanation or a line to state why these differences may have been seen? Is this intrinsic to inhibitor experiments or conditions or what?

2) Authors have some hypocotyl length experiments in dark. These experiments have one issue. To my understanding hypocotyl elongation stops when cotyledons open. When cotyledons do not open, (e.g., in dark) how can we be sure that the length measured is about hypocotyl elongation or slow growth? In other words, if you keep them long enough, would *tot3* mutants grow as much as *Col-0*?

3) Thanks for doing all the TWO-way ANOVA tests. However, Tukey's test still carries out pair-wise differences and not really G x E. To test for the effect of G x E, in the Two way ANOVA, one needs to include the "interaction" as a term and test whether there is a significant effect (in many of their data, there is.). G x E tests whether the response of two genotypes (i.e., slope of the reaction norm for example is parallel or differs) to conditions is different. Wherever required, please carry out this properly.

Reviewer #2 (Remarks to the Author):

I very much appreciate the authors efforts that fully responded to my comments. Newly provided data significantly improved the manuscript, and I really enjoyed reading the revised manuscript.

I only have following minor comments.

1. line 38 in Abstract: May add 'in *Arabidopsis thaliana*' to be more precise and informative.

2. line 155 and later, also in fig legend: Typo for 'homologues'

3. line 193-194: Based on presented data, not true for IAA gene expression. You do see the significant change in the mutant.

4. line 236: Did not find the data in Sup Fig.16a

5. line 237: Fig.4a,c supposed to be Fig.5a,c?

6. line 238: TOT3L4 and TOT3L5 phosphosites are not marked in bold as explained in Sup. Data S6B

7. line 283-284: Since elongation of the mutant at 28 degree it compromised, I'm not sure if it is correct to interpret/describe as in this way.

8. line 311: May tone down. 'indicate' to 'suggest'

9. line 386-387: Should remove 'reasonable' for providing the data used in this study.

We thank both reviewers for the positive feedback on the revised manuscript. We have addressed the remaining minor comments and highlighted these changes in green in the text.

Reviewer #1 (Remarks to the Author):

1) There are differences with previously published data on inhibiting the BRZ signaling (with ref 41). Authors report a growth promotive effect at lower temperatures and growth inhibitory effect at higher temperatures. While the authors state that "in their hands" this is what they see, both can not be true. It would be better if the authors confirm and provide a potential explanation or a line to state why these differences may have been seen? Is this intrinsic to inhibitor experiments or conditions or what?

REPLY: We would like to point out the differences in experimental conditions between our setup and reference [41]. In our setup, we used a purer brassinolide (>95%, OlchemIm), instead of epibrassinolide (>=85%, Sigma Aldrich) in reference [41]. This may result in a difference in the activity of brassinolides. However, we used a lower concentration of BL (50 nM) in contrast to 100 nM in the reference [41]. Further, we also used the short day (16h dark/8 h light) set up throughout our manuscript. In reference [41], the long day light regime (16 h light/8 h dark) was used and all the plants were grown at 20 °C for 3 days first before being separated and transferred to different temperature for another 4 days. In our setup, after germination, plants were put immediately at different temperatures to maximize the difference between the treatments and genotypes. These multiple factors may therefore contribute to the differences between our manuscript and the previously published work. We have added a short sentence to address this in our manuscript: "This can be due to differences in the experimental setup and/or the different source or concentration of brassinolide (see Methods).", and added details on the hormone source in the Supplementary Methods section. In addition, both growth inhibiting and growth repressing effects of a hormone are also observed for auxin, so there might be a concentration or condition-dependent regulation. We also added "..., but both growth promoting and repressing activities are not uncommon for a hormone⁴⁴".

2) Authors have some hypocotyl length experiments in dark. These experiments have one issue. To my understanding hypocotyl elongation stops when cotyledons open. When cotyledons do not open, (e.g., in dark) how can we be sure that the length measured is about hypocotyl elongation or slow growth? In other words, if you keep them long enough, would tot3 mutants grow as much as Col-0?

REPLY: We previously performed a dark experiment and measured the hypocotyl length at 6 DAG (see new Supplementary Figure S15d). The difference in response to high temperature between Col-0 and *tot3* mutants is comparable to the data at 3 DAG. For practical reasons, we therefore performed the follow-up experiments using the 3 DAG treatment since the early time point is already sufficient to observe the differences. Obviously, we cannot completely rule out that there might be differences when growing longer, but up to 6 DAG the hypocotyl length difference stands. Nevertheless, to avoid confusion between “elongation”, “growth” and “length” we have modified the text where relevant.

3) Thanks for doing all the TWO-way ANOVA tests. However, Tukey's test still carries out pair-wise differences and not really G x E. To test for the effect of G x E, in the Two way ANOVA, one needs to include the "interaction" as a term and test whether there is a significant effect (in many of their data, there is..). G x E tests whether the response of two genotypes (i.e., slope of the reaction norm for example is parallel or differs) to conditions is different. Wherever required, please carry out this properly.

REPLY: We apologize that this was not clear in the revised version. The requested information was available in the Source Data Tables, where we added all details on the two-way ANOVA results, including information on interaction. We now also included this information directly on the relevant Figures.

Reviewer #2 (Remarks to the Author):

1. line 38 in Abstract: May add 'in *Arabidopsis thaliana*' to be more precise and informative.

REPLY: We have added this as suggested.

2. line 155 and later, also in fig legend: Typo for 'homologues'

REPLY: We would like to clarify that in the context of polyploid plants, such as wheat, homeologues (or alternatively written as homoeologues) are “pairs of genes that originated by speciation and were brought back together in the same genome by allopolyploidization” (Glover et al, 2016, Trends in Plant Sci, doi: 10.1016/j.tplants.2016.02.005). This is different from homologues which are only simply pairs of related genes originated from a common ancestry. The use of “homeologous” or “homeologues” is thus not a typo.

3. line 193-194: Based on presented data, not true for IAA gene expression. You do see the significant change in the mutant.

REPLY: Visually, we agree with the reviewer that *IAA29* expression seems to be affected in *tot3-2* mutant. However, the two-way ANOVA test shows that the interaction “genotype x temperature” is insignificant (p-value = 0.08, see Source data), which supports that the response to the temperature cues is similar for both genotypes. Therefore, we have interpreted this as such in the text. We have now rephrased this as follows: “The observation that transcript levels of *YUC8* (a rate limiting enzyme in warm temperature-induced auxin biosynthesis) and of *IAA29* (an auxin response gene involved in hypocotyl growth) were not or hardly affected in the *tot3-2* mutant background (Fig. 4e and Supplementary Fig. 15a) suggests that auxin responses are largely not affected by *TOT3*.”

4. line 236: Did not find the data in Sup Fig.16a.

REPLY: We have corrected this mistake. The text should have referred to Figure 5a, c. On line 242, Supplementary Figure 16b is corrected to Supplementary Figure 16.

5. line 237: Fig.4a,c supposed to be Fig.5a,c?

REPLY: We have corrected this.

6. line 238: TOT3L4 and TOT3L5 phosphosites are not marked in bold as explained in Sup. Data S6B

REPLY: We have now marked TOI4 and TOI5 phosphosites in bold.

7. line 283-284: Since elongation of the mutant at 28 degree it compromised, I'm not sure if it is correct to interpret/describe as in this way.

REPLY: We have modified the text as follows: "Treatment with brassinolide increases hypocotyl length at 21 °C similarly in Col-0 and *tot3-2* (Fig. 6b). In our hands, however, the same brassinolide treatment resulted in a shorter hypocotyl at 28 °C (Fig. 6b), in contrast to previously reported results⁴¹. This can be due to differences in the experimental setup and/or the different sources of brassinolide (see Methods), but both growth promoting and repressing activities are not uncommon for a hormone⁴⁴. Nevertheless, a strong reduction in hypocotyl length in the presence of brassinolide at 28 °C was not observed in *tot3-2* plants (Fig. 6b), suggesting that TOT3 might impact brassinosteroid signalling."

8. line 311: May tone down. 'indicate' to 'suggest'

REPLY: We have rephrased this as suggested.

9. line 386-387: Should remove 'reasonable' for providing the data used in this study.

REPLY: We have corrected this.